# Dragging with Geometry: From Pixels to Geometry-Guided Image Editing

**Xinyu Pu**[1]     **Hongsong Wang**[1,*]     **Jie Gui**[1,2,*]     **Pan Zhou**[3]
[1]Southeast University   [2]Purple Mountain Laboratories   [3]Singapore Management University
{xinyupu,hongsongwang,guijie}@seu.edu.cn   panzhou3@gmail.com

## Abstract

Interactive point-based image editing serves as a controllable editor, enabling precise and flexible manipulation of image content. However, most drag-based methods operate primarily on the 2D pixel plane with limited use of 3D cues. As a result, they often produce imprecise and inconsistent edits, particularly in geometry-intensive scenarios such as rotations and perspective transformations. To address these limitations, we propose a novel geometry-guided drag-based image editing method—GeoDrag, which addresses three key challenges: 1) incorporating 3D geometric cues into pixel-level editing, 2) mitigating discontinuities caused by geometry-only guidance, and 3) resolving conflicts arising from multi-point dragging. Built upon a unified displacement field that jointly encodes 3D geometry and 2D spatial priors, GeoDrag enables coherent, high-fidelity, and structure-consistent editing in a single forward pass. In addition, a conflict-free partitioning strategy is introduced to isolate editing regions, effectively preventing interference and ensuring consistency. Extensive experiments across various editing scenarios validate the effectiveness of our method, showing superior precision, structural consistency, and reliable multi-point editability. Project page: https://xinyu-pu.github.io/projects/geodrag.

## 1 Introduction

Image editing (Cho et al., 2024; Mokady et al., 2022; Shi et al., 2024) has seen remarkable progress in recent years, largely driven by the emergence of powerful generative models (Hertz et al., 2022; Kawar et al., 2023; Mokady et al., 2022; Yang et al., 2024b; Zhou et al., 2025). Among these, text-guided image editing (Ruiz et al., 2023; Huberman-Spiegelglas et al., 2024; Zhou et al., 2025; Nguyen et al., 2024) has become a widely adopted paradigm, allowing users to modify images using natural language prompts. While expressive and flexible, this approach often falls short in providing fine-grained spatial control—especially when precise, localized, or geometry-sensitive edits are required (Pan et al., 2023; Shi et al., 2024; Liu et al., 2024; Ling et al., 2024; Hou et al., 2024; Zhang et al., 2025; Chen et al., 2024).

To overcome these limitations, point-based image editing (Pan et al., 2023; Shi et al., 2024; Liu et al., 2024; Ling et al., 2024; Hou et al., 2024; Zhang et al., 2025; Chen et al., 2024) has emerged as a powerful alternative, gaining traction for its user-friendly, precise manipulation. By enabling users to specify handle-to-target point pairs, methods such as DragGAN (Pan et al., 2023) and its successors (Shi et al., 2024; Liu et al., 2024; Ling et al., 2024; Hou et al., 2024; Zhang et al., 2025; Chen et al., 2024; Shin et al., 2024) offer intuitive and precise image manipulation. They often adopt a two-step pipeline: (1) motion supervision, dragging the handle point toward the target, and (2) point tracking, monitoring the handle's updated position during editing.

**Motivation.** Despite their success, existing point-based methods often rely on iterative gradient-based optimization, which is computationally intensive and impractical for real-time use. To improve efficiency, recent approaches like FastDrag (Zhao et al., 2024) and RegionDrag (Lu et al., 2024) propose one-step editing via latent relocation based on dense displacement fields constructed over user-specified regions. However, these methods operate purely in the 2D pixel plane, and ignore

---

*Corresponding authors. Code is available at https://github.com/xinyu-pu/GeoDrag

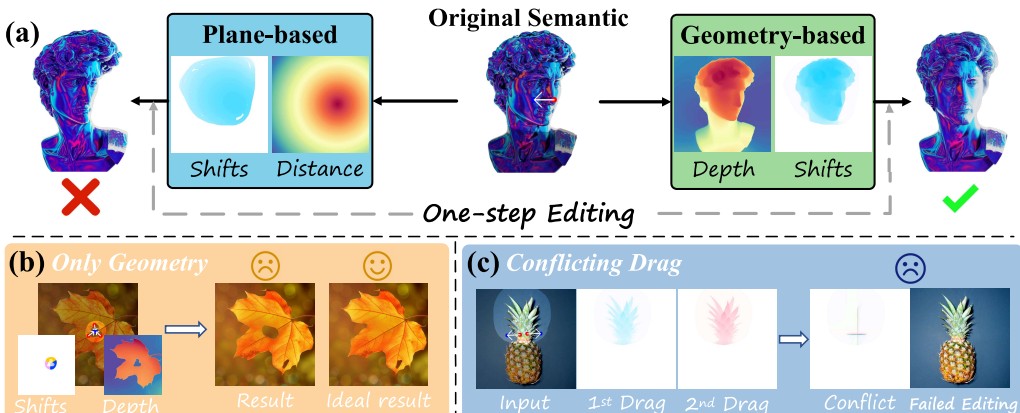

Figure 1: **(a)** Displacement fields. Left: Plane-based estimation (e.g., FastDrag (Zhao et al., 2024)) lacks geometric awareness and introduces structural inconsistencies under geometry-intensive edits like rotation. Right: Our geometry-aware GeoDrag aligns with 3D structure—near pixels move more, far pixels less. **(b)** Relying solely on 3D geometry can produce discontinuous displacements near object boundaries. **(c)** Two nearby handle-target pairs with opposing directions (e.g., leftward and rightward drags) may conflict, causing displacement cancellation and editing failure. The color legend (Shift/Distance/Depth) is provided in Appendix H.

the underlying 3D scene geometry. This becomes a critical limitation for complex transformation or geometry-intensive edits—like rotations or perspective shifts—where 2D-only reasoning leads to structural artifacts, unnatural deformations, and spatial misalignment. For example, plane-aware strategy decays displacement strength based solely on pixel distance, and ignores 3D geometry, resulting in perceptual distortions of human's face as shown in Fig. 1(a, left).

To improve the realism and semantic consistency of image editing, incorporating 3D geometric information is essential, as it offers richer structural cues beyond the 2D pixel plane. However, this introduces three key challenges. 1) **How can geometry be integrated into pixel-level editing?** While 3D cues (e.g., depth maps) are informative, they do not align directly with pixel-wise operations. For example, a drag defined in the image plane may become ambiguous in 3D due to perspective and depth variation—highlighting the need for a mechanism to incorporate geometry into 2D editing pipelines. 2) **Is geometry alone sufficient for high-quality editing?** Although helpful for preserving global structure, geometry alone may cause issues. Displacement fields based solely on 3D geometry (e.g., depth) often become discontinuous near object boundaries (see Sec. 3.1), disrupting the diffusion process and causing semantic artifacts as shown in Fig. 1(b). 3) **How to reconcile guidance from multiple handle-target pairs?** In real scenarios, users often specify multiple drag points. If their displacement fields overlap—especially with opposing directions—they can destructively interfere, even with distance-based weighting as in FastDrag (Zhao et al., 2024). This leads to displacement cancellation and failed edits as illustrated by Fig. 1(c). These challenges call for a unified framework that integrates 3D geometry and 2D cues while resolving conflicts for precise and coherent manipulation.

**Contributions.** To tackle these challenges, we propose GeoDrag—a drag-based image editing framework that is both geometry-aware and plane-aware, ensuring coherent, high-fidelity, and fast one-step image manipulation. Built upon the latent consistency model (LCM) (Luo et al., 2023), GeoDrag predicts a dense displacement field directly in the noisy latent space at a specific diffusion timestep—circumventing iterative optimization and enabling fast and efficient editing. GeoDrag introduces three key innovations to resolve the above three challenges, respectively. 1) **Geometry-aware field modeling**: To resolve the mismatch between 3D geometry and 2D editing, GeoDrag introduces a novel influence function that modulates displacement strength based on 3D geometric relationships. By leveraging depth contrast, it ensures that nearby regions undergo stronger projective motion, while distant areas move more subtly—preserving 3D structure (see the well-edited human face in Fig. 1(a, right)). 2) **Spatial plane modulation**: Addressing the limitations of using 3D geometry alone, GeoDrag incorporates a spatial influence function based on 2D pixel plane. This improves local structure preservation and editing precision, especially in flat or geometry-ambiguous

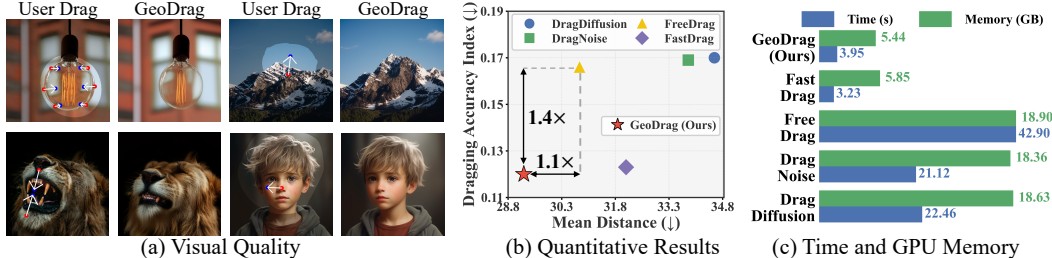

(a) Visual Quality  (b) Quantitative Results  (c) Time and GPU Memory

Figure 2: Experimental Comparison. (a) Representative edits across diverse scenarios. (b) Quantitative results on DragBench: lower MD and DAI indicate more accurate editing. (c) Runtime and memory comparison across our GeoDrag and previous SoTAs.

regions. 3) **Conflict-Free Partitioning**: To mitigate conflicts in multi-point editing, GeoDrag segments the editing mask into non-overlapping sub-regions, each associated with its nearest handle point. Independent displacement fields are computed per region, avoiding destructive interference and ensuring coherent multi-point manipulation. Together, these contributions allow GeoDrag to perform fast, high-quality, and semantically consistent edits in a single step—advancing the state of controllable, geometry-aware image manipulation.

Extensive experiments verify the effectiveness and efficiency of GeoDrag. As shown in Fig. 2, we provide a three-part visualization: (a) Visual Quality – GeoDrag delivers high-quality edits across challenging scenarios, including multi-point editing (e.g., bulb), structure-preserving deformation (e.g., mountain), localized manipulation (e.g., lion's mouth), and geometry-aware tasks (e.g., face rotation); (b) Quantitative Results – GeoDrag achieves superior alignment accuracy, improving runner-up's dragging accuracy index metric (DAI) by 1.4x and mean distance metric (MD) by 1.1x; (c) Efficiency – GeoDrag offers a favorable trade-off between speed and memory. While its runtime gain is modest, it remains lightweight and highly competitive due to its strong editing performance.

## 2  RELATED WORK

**Text-Based Image Editing.** Text-based image editing manipulates images via natural language prompts. Gal et al. (2023) utilize textual inversion for personalized generation by embedding user-specific concepts. DiffusionCLIP (Kim et al., 2022) fine-tunes diffusion models with CLIP supervision, while Prompt-to-Prompt (Hertz et al., 2022) achieves train-free editing by modifying cross-attention maps. Null-text inversion (Mokady et al., 2022) optimizes unconditional text embeddings for faithful reconstruction of real images. Imagic (Kawar et al., 2023) interpolates between text and image-specific embeddings but requires per-task tuning. Other approaches like CycleDiffusion (Wu & la Torre, 2023) and DDPM inversion (Huberman-Spiegelglas et al., 2024) explore latent spaces to support high-quality editing. InstructPix2Pix (Brooks et al., 2023) trains on instruction-image pairs, enabling direct prompt-driven editing. RPG (Yang et al., 2024b) introduces a multimodal LLM for reasoning and planning editing. Despite their flexibility, text-based methods often lack spatial precision and fine-grained control, limiting their applicability for detailed editing tasks.

**Interactive Point-Based Image Editing.** Point-based methods enable precise manipulation by directly dragging image elements. DragGAN (Pan et al., 2023) introduced this paradigm with GANs, later improved by diffusion-based approaches (Ho et al., 2020; Rombach et al., 2022; Luo et al., 2023) like DragDiffusion (Shi et al., 2024), which integrates motion supervision and identity-preserving fine-tuning. Extensions enhance usability (EasyDrag (Hou et al., 2024)), stability (FreeDrag (Ling et al., 2024)), semantic control (DragNoise (Liu et al., 2024)), or robustness (Good-Drag (Zhang et al., 2025), StableDrag (Cui et al., 2024)). To boost efficiency, FastDrag (Zhao et al., 2024), RegionDrag (Lu et al., 2024), and SDEDrag (Nie et al., 2024) use lightweight latent manipulations, while DragonDiffusion (Mou et al., 2024b), DiffEditor (Mou et al., 2024a), and Instant-Drag (Shin et al., 2024) introduce energy-based or real-time formulations. Despite progress, they remain limited to 2D pixel reasoning, restricting realism in geometry-sensitive edits. We address this with **GeoDrag** for controllable, structure-preserving, and efficient image editing. In parallel, FlowDrag (Koo et al., 2025) uses mesh reconstruction and iterative deformation to improve editing

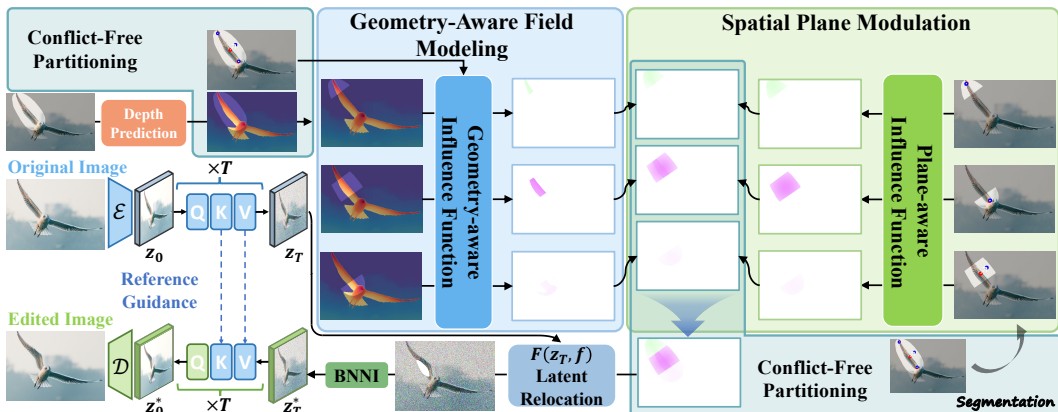

Figure 3: Overall framework of GeoDrag. In drag pipeline, the mask is split into sub-regions, each with a pair of drag points. For each sub-region, the geometry- and plane-aware displacement fields are independently calculated (see Sec. 3.1 and Sec. 3.2). Subsequently, these fused fields are aggregated without conflict (see Sec. 3.3). The final field enables one-step editing via latent relocation and interpolation, with reference guidance to preserve semantics.

quality, but at a higher computational cost, that limits its responsiveness. By contrast, GeoDrag achieves geometry-consistent control while remaining responsive for interactive editing.

## 3 METHODOLOGY

We begin by formalizing the task of interest. Given an image and a set of $k$ point pairs $\{(\boldsymbol{h}_i, \boldsymbol{t}_i)\}_{i=1}^{k}$, where each $\boldsymbol{h}_i$ is a handle point and $\boldsymbol{t}_i$ its corresponding target, the goal is to move each $\boldsymbol{h}_i$ toward $\boldsymbol{t}_i$ while preserving semantic consistency and visual realism. Recent efficient methods such as FastDrag (Zhao et al., 2024) and RegionDrag (Lu et al., 2024) enable a fast one-step editing by computing a displacement field $\boldsymbol{f} \in \mathbb{R}^{H \times W \times 2}$, which warps the latent $\boldsymbol{z}_T$ at timestep $T$ via forward mapping:

$$\boldsymbol{z}_{T(i+f_{i,j,1}, j+f_{i,j,0})}^{*} = \boldsymbol{z}_{T(i,j)}. \tag{1}$$

Each $\boldsymbol{f}_{i,j} = (f_{i,j,1}, f_{i,j,0})$ defines how much the latent feature at spatial location $(i,j)$ should be shifted along the x and y axes in the image. Next, the modified latent $\boldsymbol{z}_T^{*}$ is then passed into a well-trained diffusion model to generate the edited image.

Existing methods (Zhao et al., 2024; Lu et al., 2024; Nie et al., 2024) construct $\boldsymbol{f}$ using only 2D pixel-plane heuristics, ignoring the underlying 3D structure. This estimated $\boldsymbol{f}$ often leads to unrealistic deformations, semantic breaks, and perspective issues—especially in geometry-sensitive edits like rotations or viewpoint shifts (see Fig. 1 (a)). Additionally, multiple handle-target pairs can create overlapping and conflicting displacement fields, causing inconsistent guidance and editing failures (see Fig. 1 (c)).

To address these issues, we propose GeoDrag which jointly leverages 3D geometry and 2D spatial priors to produce accurate and coherent displacement fields. As illustrated in Fig. 3, GeoDrag consists of three components: (1) a geometry-aware field modeling module that adjusts motion strength using depth cues; (2) a spatial plane modulation that combines 3D and 2D guidance; and (3) a conflict-free partitioning that decomposes the editing mask to resolve conflicting drag signals. These components respectively resolve three fundamental challenges associated with geometry-aware drag-based editing, and they together enable high-quality and geometry-consistent image edits in a single step. Moreover, to avoid the over-smoothing often introduced by interpolation, we further refine the interpolated latents with a masked stochastic DDIM update, which injects randomness only inside the interpolated region while keeping the rest deterministic. Formally, given a binary mask $\boldsymbol{M}$ indicating the interpolated area, the sample step is

$$\boldsymbol{z}_{t-1}^{*} = \sqrt{\bar{\alpha}_{t-1}} \hat{\boldsymbol{z}}_0^{*} + \sqrt{1 - \bar{\alpha}_{t-1} - \sigma_t^2} \odot \boldsymbol{M} \, \epsilon_\theta(\boldsymbol{z}_t^{*}, t) + \sigma_t \, (\epsilon \odot \boldsymbol{M}). \tag{2}$$

This post-interpolation refinement preserves global coherence, effectively alleviating blur without incurring extra sampling overhead. Below, we elaborate on these three components and their resolved challenges in turn. Specifically, we focus on the case of a single handle-target pair in Sec. 3.1 and Sec. 3.2, and extend the approach to multi-point editing in Sec. 3.3.

## 3.1 GEOMETRY-AWARE FIELD MODELING

Incorporating 3D geometry into 2D image editing poses a key challenge: pixel-level operations on the image plane do not directly correspond to transformations in 3D space. For instance, applying the same 3D displacement to different points can lead to inconsistent 2D motions due to perspective distortion and depth variation. To bridge this gap, our core idea is to project 3D displacements into the image plane while preserving 3D structure information. Accordingly, we design a geometry-aware influence function that converts 3D drag displacements into pixel plane by incorporating relative depth between pixels and the handle point. This strategy ensures that pixels respond to the displacement in a depth-consistent manner: pixels with lower depth to handle point are influenced more, while those at larger depths move less. By aligning the 2D displacement strength with 3D proximity, the method preserves the consistency of the 3D structure during 2D "dragging", avoiding spatial tearing or inconsistent deformation on the image plane.

Specifically, consider a drag operation applied to 3D space $(x, y, z)$ with a 3D displacement vector $(\delta x, \delta y, \delta z)$. Given a camera intrinsic $\mathbf{K}$ which in unknown in this work, the corresponding projected 2D coordinate $(u, v)$ on the image plane can be computed as follows:

$$z \begin{bmatrix} u \\ v \\ 1 \end{bmatrix} = \mathbf{K} \begin{bmatrix} x \\ y \\ z \end{bmatrix} = \begin{bmatrix} f_x & 0 & c_x \\ 0 & f_y & c_y \\ 0 & 0 & 1 \end{bmatrix} \begin{bmatrix} x \\ y \\ z \end{bmatrix}. \tag{3}$$

Applying a small 3D displacement $(\delta x, \delta y, \delta z)$, its projected 2D shift $(\delta u, \delta v)$ becomes:

$$\delta u = f_x \left( \frac{x + \delta x}{z + \delta z} - \frac{x}{z} \right), \quad \delta v = f_y \left( \frac{y + \delta y}{z + \delta z} - \frac{y}{z} \right), \tag{4}$$

where $f_x$ and $f_y$ are the focal lengths of the camera. Since the drag operations are defined on the 2D image plane (in our task), the motion along the optical axis (i.e., the $z$-axis) can be reasonably neglected. Thus, Eq. (4) can be simplified as:

$$\delta u = f_x(\delta x/z), \quad \delta v = f_y(\delta y/z). \tag{5}$$

Furthermore, consider another arbitrary 3D point $(x', y', z')$ which is subjected to the same displacement vector $(\delta x, \delta y, \delta z)$. We can also compute its corresponding 2D displacement $(\delta u', \delta v')$ as $\delta u' = f_x \frac{\delta x}{z'}$ and $\delta v' = f_y \frac{\delta y}{z'}$. In this way, combining Eq. (5) yields

$$\delta u' = (z/z')\delta u, \quad \delta v' = (z/z')\delta v. \tag{6}$$

Eq. (6) implies that $(u', v')$ with a smaller depth (closer to the camera) exhibit a greater pixel-plane displacement due to the inverse proportionality between displacement and depth. Based on this observation, the geometry-aware field can be constructed as follows:

$$\boldsymbol{f}_d = (\boldsymbol{\zeta_h}/\boldsymbol{\zeta})^\alpha \cdot \boldsymbol{d} = (\boldsymbol{\zeta_h}/\boldsymbol{\zeta})^\alpha \cdot (\boldsymbol{t} - \boldsymbol{h}), \tag{7}$$

where $\boldsymbol{\zeta}$ denotes the depth map within mask, and $\boldsymbol{\zeta_h}$ is the depth of handle point $\boldsymbol{h}$. The scalar $\alpha$ serves as a modulation factor, controlling the sensitivity of displacement scaling to depth variations. $\boldsymbol{d}$ is the drag direction from handle $\boldsymbol{h}$ to target $\boldsymbol{t}$. The geometry-aware influence function (i.e., Eq. (7)) calculates the geometry-aware displacement field (as shown in the central blue-shaded region of Fig. 3) based on the underlying 3D structure. By incorporating depth-dependent modulation, GeoDrag brings 3D geometric information into 2D drag editing, enabling structure-preserving manipulations. Our strategy resolves the challenge by maintaining geometric consistency between the perceived 2D deformation and the actual 3D transformation.

## 3.2 SPATIAL PLANE MODULATION

While geometry-aware displacement field provides structural consistency by incorporating 3D depth information, it alone is insufficient for producing high-quality edits—particularly in regions with fine

details or near object boundaries. This limitation arises because geometry-aware motion distributes influence uniformly in 3D space, making it less responsive to subtle and local deformations in the 2D image plane (see Fig. 1(b)). This is the second major challenge discussed in Sec. 1. To overcome this, we propose a spatial plane modulation strategy that complements the global structure-preserving behavior of the geometry-aware field with local and pixel-level controllability. This hybrid approach enables precise and sharp edits while retaining geometric coherence.

Our fusion mechanism draws inspiration from elastic force propagation: deformation peaks at the force point and decays with distance. Mimicking this behavior, we define a plane-aware field that decays spatially from the handle point. This formulation allows localized and responsive editing, especially near fine structural features. While this idea is conceptually similar to the influence decay used in FastDrag (Zhao et al., 2024), our method avoids plane geometric constructions like similar triangles and instead adopts a simpler, vectorized formulation that is more computationally efficient and easier to integrate.

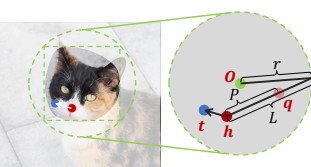

Given a handle point $h$ and its target point $t$, the drag vector is $d = t - h$. The displacement at each pixel is then computed by modulating the displacement vector $d$ according to its spatial distance to $h$, following the spatial influence function:

$$f_p = \left(1 - (P/L)^\beta\right) \cdot d, \qquad (8)$$

Figure 4: Illustration of $L$ in the plane-aware field. $O$ and $r$ are the center of the outer circle and radius, respectively.

where $1 \in \mathbb{R}^{h \times w}$ denotes a matrix with all elements as one, $P \in \mathbb{R}^{h \times w}$ denotes the Euclidean distance from each pixel to the handle point $h$, $L \in \mathbb{R}^{h \times w}$ is the maximum propagation distance along the ray from $h$ to each pixel, and $\beta$ controls how sharply the influence falls off with distance.

The propagation is restricted within a circular region enclosing the editing mask, ensuring the influence fades smoothly near the mask boundary. As shown in Fig. 4, to compute $L$, we solve for the ray-circle intersection:

$$|h + tv - O|^2 = r^2 \quad \Rightarrow \quad t = -v \cdot (O - h) + \sqrt{(v \cdot (O - h))^2 - (|O - h|^2 - r^2)}, \qquad (9)$$

where $v = \frac{q-h}{|q-h|}$ is the unit direction vector from $h$ to pixel $q$, and $L = |h + tv|$ gives the maximum extent of influence.

To achieve both structural consistency and local flexibility, we fuse the geometry-aware field $f_d$ and the plane-aware field $f_p$ into a single displacement field $f$:

$$f = (1 - \lambda) \cdot f_p + \lambda \cdot f_d. \qquad (10)$$

Here, $\lambda$ is a spatially adaptive fusion weight based on the distance from each pixel to handle point, formulated as $\lambda = P/(P + \gamma)$. Where $\gamma \geq 0$ is a hyperparameter controlling the balance between global geometry-aware and local plane-aware influence. A smaller $\gamma$ favors geometric consistency, while a larger $\gamma$ increases responsiveness to localized changes. As shown in the rightmost part of Fig. 3, the fused field integrates 3D geometric priors and 2D plane cues to enable semantically coherent and structure-preserving displacements. Since the ideal fusion scale varies across different object sizes and editing regions, we define $\gamma$ as a scalar multiple of the diameter of the enclosing mask circle, making the fusion strategy adaptive to the editing context.

### 3.3 CONFLICT-FREE PARTITIONING

When multiple handle-target pairs are involved in drag-based editing, directly aggregating their displacement fields can lead to destructive interference—particularly when nearby handles induce conflicting motion directions. This is the third major challenge outlined in Sec. 1. Such interference often results in weakened motion strength, ambiguous displacement patterns, and ultimately, failed or unintuitive edits. Naive approaches like distance-based weighting are insufficient in these scenarios since they cannot fully decouple the influence of closely spaced or competing drag handles (see Fig. 6). To resolve this, we introduce a conflict-free partitioning that enforces local independence by spatially partitioning the editing mask into multiple sub-regions. Each sub-region is exclusively influenced by a single handle point, ensuring that conflicting displacements are isolated and handled separately. This strategy significantly improves editing precision and prevents cross-handle interference.

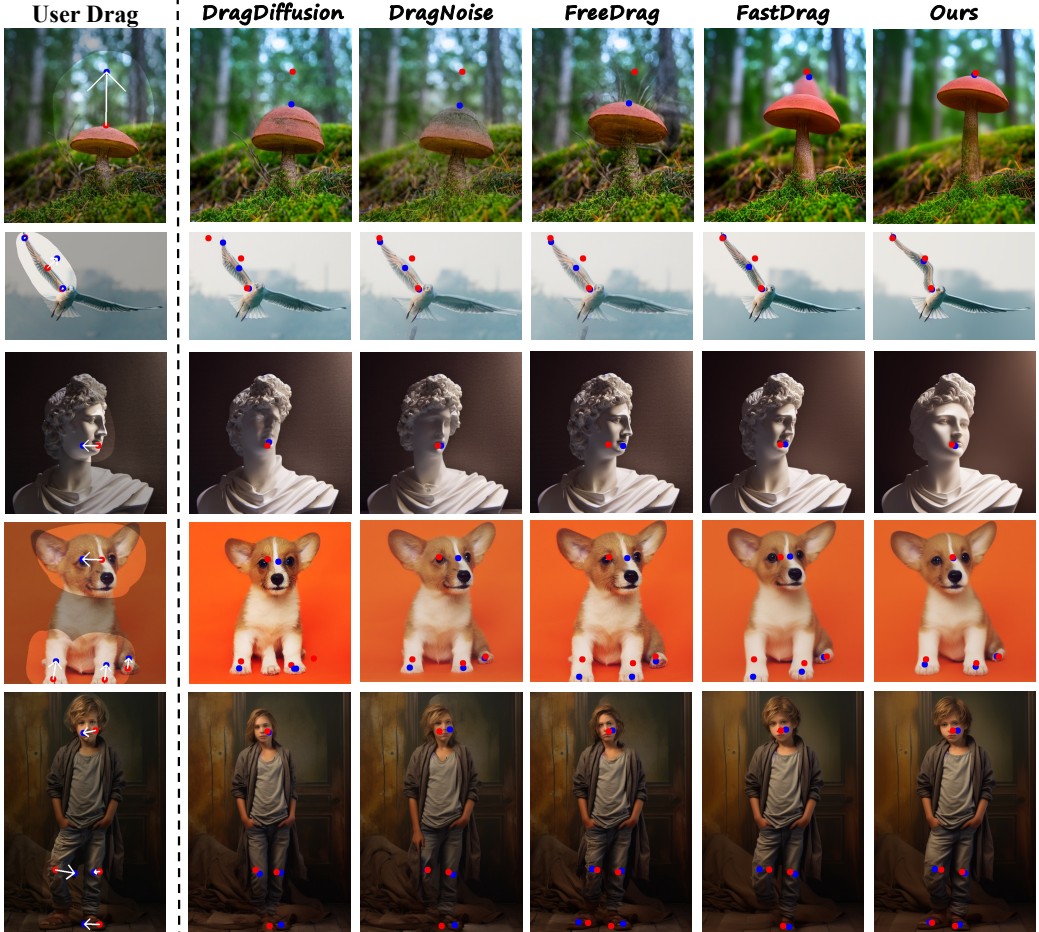

Figure 5: Qualitative comparisons with the state-of-the-art interactive point-based methods. See Appendix D for extended qualitative comparisons, including additional visualizations in Fig. 23. In addition, see Appendix G for details on how mask selection influences editing behavior. Red points mark handles, and blue points mark targets; the same applies to the following figures.

Given an editing mask $M$, we divides it into disjoint sub-regions $\mathcal{S}_i$ via Eq. (11), where each pixel $q \in M$ is assigned to the nearest handle point $h_i$.

$$\mathcal{S}_i = \left\{ q \in M \mid i = \arg\min_{j \in \{1,\ldots,N\}} \|q - h_j\|_2 \right\}. \tag{11}$$

This Voronoi-like partition ensures that each sub-region $\mathcal{S}_i$ is controlled only by its corresponding handle point $h_i$, effectively decoupling the influence zones and eliminating destructive overlap.

Once the partitioning is established, we compute the displacement field $f_i$ independently for each sub-region $\mathcal{S}_i$ using the hybrid geometry- and plane-aware formulation from the previous section. The final displacement field $f$ is constructed by assigning the corresponding sub-field to each pixel:

$$f(q) = f_i(q), \quad \text{for } q \in \mathcal{S}_i. \tag{12}$$

This region-wise aggregation ensures that each pixel is influenced by only one handle, avoiding directional conflicts and enabling precise and localized edits—even when multiple drags are applied. Despite being a hard partition, our ablations (see Fig. 6) show it performs better than soft partitioning (Directly Add, Pixel Distance, and Drag Magnitude).

Table 1: Quantitative results on DRAGBENCH. Lower **MD** and **DAI** indicate higher editing precision, and higher **IF** reflects greater similarity between original and edited images. **Time** is the average editing time per point, and **Mem** is the peak GPU memory (GB).

| Approach | MD ↓ | DAI$_1$ ↓ | DAI$_{10}$ ↓ | DAI$_{20}$ ↓ | IF ↑ | Preparation | Time (s) | Mem |
|---|---|---|---|---|---|---|---|---|
| DragDiffusion (Shi et al., 2024) | 34.57 | 0.181 | 0.170 | 0.160 | 0.871 | ~1 min (LoRA) | 22.46 | 18.63 |
| FreeDrag (Ling et al., 2024) | 30.80 | 0.183 | 0.166 | 0.151 | 0.845 | ~1 min (LoRA) | 42.90 | 18.90 |
| CLIPDrag (Jiang et al., 2024) | 34.62 | 0.195 | 0.174 | 0.158 | **0.891** | ~1 min (LoRA) | 38.21 | 22.72 |
| AdaptiveDrag (Chen et al., 2024) | 32.38 | 0.180 | 0.154 | 0.146 | 0.830 | ~1 min (LoRA) | 46.30 | 7.71 |
| DragNoise (Liu et al., 2024) | 33.84 | 0.179 | 0.169 | 0.158 | 0.861 | ~1 min (LoRA) | 21.12 | 18.36 |
| FastDrag (Zhao et al., 2024) | 32.10 | 0.131 | 0.123 | 0.115 | 0.850 | ✗ | **3.23** | 5.85 |
| **GeoDrag (Ours)** | **29.24** | **0.128** | **0.120** | **0.111** | 0.847 | ✗ | 3.95 | **5.44** |

| User Drag | RAD | Directly Add | Pixel Distance | Drag Magnitude |
|---|---|---|---|---|

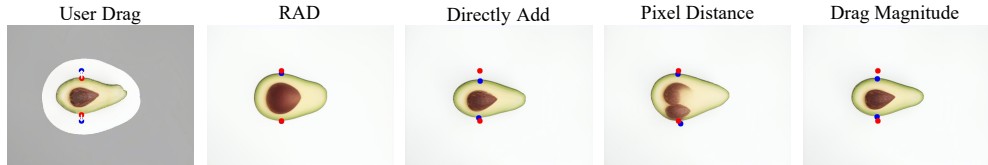

Figure 6: Ablation study on multi-point drag strategies. See quantitative results in Fig. 10(a).

## 4 EXPERIMENTS

### 4.1 QUALITATIVE EVALUATION

We conduct qualitative comparisons against existing state-of-the-art drag-based image editing methods, including DragDiffusion (Shi et al., 2024), DragNoise (Liu et al., 2024), FreeDrag (Ling et al., 2024), and FastDrag (Zhao et al., 2024). As shown in Fig. 5, GeoDrag achieves superior performance and high image quality. For instance, in the first row of Fig. 5, GeoDrag accurately drags the handle points toward the target points, preserving both structural integrity and semantic coherence, while other methods, such as FastDrag and FreeDrag, fail to maintain precise alignment. In multi-point editing (e.g., the second, fourth, and fifth rows of Fig. 5), GeoDrag successfully reshapes the wings, adjusts postures in alignment with the user-specified editing intention. In contrast, due to conflicts among multiple drag points, other methods struggle to generate coherent deformations. Observation from the last three rows of Fig. 5, GeoDrag produces 3D structure-coherent manipulation. These advantages are derived from the integration of geometric information, which provides a displacement field toward 3D structure alignment, enabling structurally consistent editing.

### 4.2 QUANTITATIVE EVALUATION

Here we conduct quantitative evaluations to validate GeoDrag where its modulation factors $\alpha$, $\beta$, and $\gamma$ are set to 1.0. Apart from (Shi et al., 2024; Liu et al., 2024; Ling et al., 2024; Zhao et al., 2024), CLIPDrag (Jiang et al., 2024) and AdaptiveDrag (Chen et al., 2024) are included for comparison. DRAGBENCH dataset (Shi et al., 2024) is used as the benchmark, while Mean Distance (MD) (Pan et al., 2023) and Image Fidelity (IF) (Kawar et al., 2023) are metrics for evaluating the editing precision and similarity between edited and original images. Dragging Accuracy Index (DAI) (Zhang et al., 2025) measures consistency in the dragged region, with DAI$_r$ evaluating a radius-$r$ patch. We also report average editing time per point and peak GPU memory.

As reported in Table 1, GeoDrag achieves the best editing precision with the lowest MD and DAI, while maintaining competitive perceptual quality reflected by IF metric. Despite not requiring any preparation overhead, such as LoRA tuning, GeoDrag outperforms all baselines. GeoDrag edits each point in 3.95 seconds on average—faster than most diffusion-based methods—and consumes little GPU memory, making it suitable for responsive applications. More results on scalability w.r.t. the number of drag points are in Appendix F. We also conduct a user study using 10 randomly selected images, each edited by FreeDrag (Ling et al., 2024), FastDrag (Zhao et al., 2024), and our

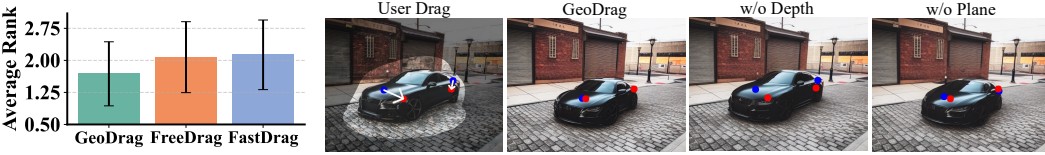

Figure 7: User study ranking on editing quality.

Figure 8: Ablation on displacement field.

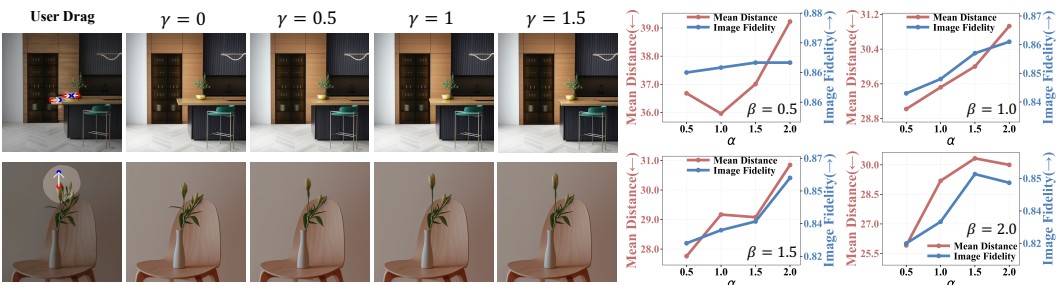

Figure 9: Hyperparameter sensitivity study. **Left**: Visual comparisons under different perceptual region $\gamma$. **Right**: Quantitative comparisons with varying modulation factors $\alpha$ and $\beta$. See more results in Appendix C.2 and Appendix C.3.

GeoDrag. 60 Participants are asked to rank the edited results (1 for best, 3 for worst). As shown in Fig. 7, GeoDrag is superior to other methods.

### 4.3 ABLATION STUDY

**Displacement Field.** We conduct ablation studies to investigate the contribution of each component in our hybrid displacement field. Specifically, **w/o Depth** removes the depth-aware field, while **w/o Plane** removes the plane-aware field. As shown in Fig. 8, removing the depth-aware field leads to inaccurate editing (e.g., failure to rotate the car). Removing plane-aware field leads to insufficient editing. The results highlight the complementary roles of 3D geometry and 2D plane prior in achieving structure-preserving and semantically coherent editing. The quantitative resuls are shown in Fig. 10(a). GeoDrag with geometry- and plane-aware displacement field achieves the best performance. Removing the geometry prior (**w/o Depth**) or the plane modulation (**w/o Plane**) leads to degradation across all metrics.

**Conflict-Free Partitioning.** We evaluate the proposed conflict-free partitioning strategy with direct summation (Directly Add), pixel-distance weighting (Pixel Distance), and drag-magnitude weighting (Drag Magnitude). The mathematical formulations of alternative strategies are provided in Appendix C.1. As shown in Fig. 6, directly adding displacement fields leads to conflict and cancellation when directions oppose; Pixel Distance and Drag Magnitude cannot effectively separate influence regions, producing duplicated and unsatisfactory results; Our method avoids interference among multiple drag points, yielding accurate results. The quantitative results are shown in Fig. 10(b). GeoDrag outperforms alternative strategies on all metrics, demonstrating the effectiveness of the conflict-free partitioning.

**Hyperparameters.** We visually analyze the influence of $\gamma$. As shown in Fig. 9(left), setting $\gamma$ between 0.5 and 1.5 strikes a good balance between geometry-aware consistency and local editability. We analyze the impact of $\alpha$ and $\beta$ in Fig. 9(right). The lower $\alpha$ indicates a smoother scale (see Eq. (7)), weakening the influence of depth contrast. This results in larger overall deformation and improved alignment. Larger $\alpha$ increases depth sensitivity, causing sharper displacement changes. This allows for finer-grained control in regions with significant depth changes. For plane-aware field, a higher $\beta$ enforces more localized and sharper deformations (see Eq. (8)), thereby better aligning the edits with user intention and resulting in lower MD. The study of the noise-scaling term $\sigma_t$ is shown in Fig. 11. Setting $\eta = 0$ leads to a fully deterministic update, often causing over-smoothed interpolations and blurred backgrounds. Increasing $\eta$ adds randomness to the interpolated

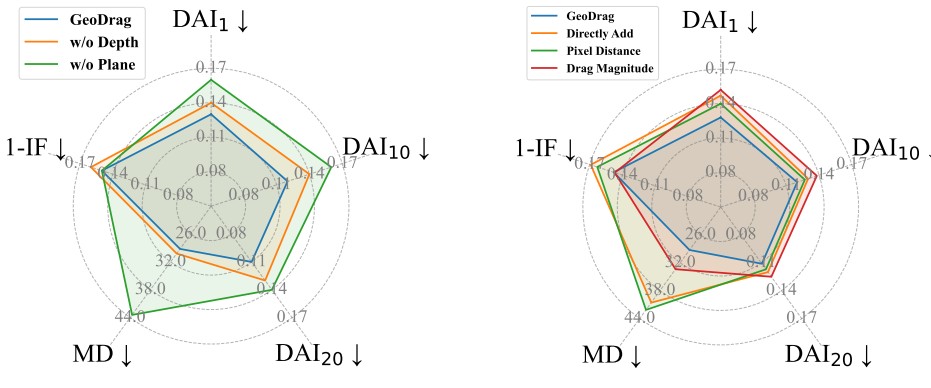

(a) Displacement field variants (cf. Fig. 8).

(b) Multi-point drag strategies (cf. Fig. 6).

Figure 10: Quantitative results of ablation study.

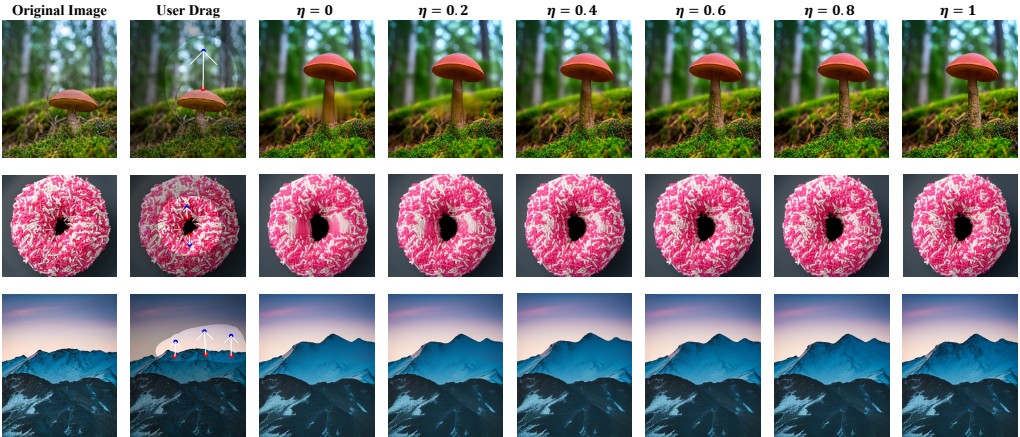

Figure 11: Ablation study on the noise-scaling term $\sigma_t$ controlled by $\eta$. We vary $\eta$ from 0 (deterministic) to 1 (stochastic).

region, helping the diffusion model better recover local details. More ablation studies and detailed analyses are provided in Appendix C.

## 5 CONCLUSION

In this paper, we propose GeoDrag, a novel interactive editing framework that integrates 3D geometric priors with 2D spatial cues. GeoDrag achieves geometry-consistent and semantically coherent image manipulation by constructing a hybrid displacement field. A geometry-aware influence function leverages depth to model 3D-consistent displacements, while a complementary plane-aware function improves the controllability of editing. To resolve multi-point conflicts, a region-aware decomposition strategy ensures conflict-free aggregation. This work offers a new perspective on how 3D geometric priors can be beneficial for precision, coherence, and controllability of 2D interactive image editing.

**Limitations.** Although GeoDrag supports one-step editing, computing multiple displacement fields introduces additional computational overhead compared to lightweight 2D-only baselines such as FastDrag (Zhao et al., 2024). But GeoDrag often yields much more geometry-consistent and high-quality image edits, e.g., 9% and 7% improvements in terms of MD and GPU memory than FastDrag as shown in Table 1.

ACKNOWLEDGMENTS

This research is supported by the National Natural Science Foundation of China (Nos.52441503 and 62302093), the Natural Science Foundation of Jiangsu Province (Nos.BK20230833), the Singapore Ministry of Education (MOE) Academic Research Fund (AcRF) Tier 1 grant (Proposal ID: 23-SIS-SMU-070). This work is supported in part by the grant of the CIPS-SMP-Zhipu Large Model Fund. We thank the Big Data Computing Center of Southeast University for providing the facility support on the numerical calculations. Any opinions, findings and conclusions or recommendations expressed in this material are those of the author(s) and do not reflect the views of the Ministry of Education, Singapore.

ETHICS STATEMENT

Image editing model may contain biases or occasionally produce sensitive or offensive outputs. Our models are presented strictly for academic and scientific research purposes. Any generated content does not reflect the personal views of the authors. Our work remains guided by a commitment to advancing AI technologies in ways that uphold ethical standards and resonate with societal values.

REPRODUCIBILITY STATEMENT

We detail the main framework of our work in Sec. 3, and provided the implementation details in Appendix B.

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

# Appendix

## A  BROADER IMPACT

GeoDrag is developed to improve the controllability and fidelity of interactive point-based image editing, enabling precise user-driven visual manipulation. The framework applies to a wide range of scenarios, including digital content creation, artistic editing, and AR/VR-based scene editing. By improving the structural consistency and semantic fidelity, GeoDrag allows users—especially non-experts—to produce high-quality and visually coherent results with minimal effort, potentially democratizing advanced visual editing workflows. However, the increased realism and control enabled by GeoDrag may raise negative societal impacts. In particular, malicious actors could leverage the system to generate visually convincing but deceptive content, contributing to disinformation, digital impersonation, or reputational harm. In addition, advancements in image editing tools increase the risk of fake imagery, potentially undermining public trust. Unethical use may also raise concerns related to individual consent and personal privacy. To address these risks, we advocate for respon-

sible deployment, transparent provenance of generated content, and further research into detection and authentication techniques.

## B  IMPLEMENTATION DETAILS

GeoDrag is implemented based on a pretrained latent diffusion model (Stable Diffusion 1.5 (Rombach et al., 2022)), and incorporates an LCM-accelerated U-Net (Luo et al., 2023) to enable efficient low-step inference. We set the number of sampling steps to 10 and use an inversion strength of $0.7$. The depth prediction model used in geometry-aware field modeling (Sec. 3.1) is Depth Anything V2 (Yang et al., 2024a). Following prior drag-based editing methods (Shi et al., 2024; Ling et al., 2024; Liu et al., 2024), we disable classifier-free guidance. For fair comparison, all baseline methods are evaluated using their default hyperparameter settings as specified in the original papers or official open-source implementations. Experiments are conducted on an RTX 4090 GPU with 24G memory.

## C  SUPPLEMENTARY ABLATION STUDY

### C.1  MATHEMATICAL FORMULATIONS OF ALTERNATIVE STRATEGIES

**Directly Add:**
$$\boldsymbol{f} = \sum_{i=1}^{k} \boldsymbol{f}_i, \tag{13}$$

**Pixel Distance:**
$$\boldsymbol{f} = \sum_{i=1}^{k} \frac{1/\boldsymbol{P}_i}{\sum_{i=1}^{k} 1/\boldsymbol{P}_i} \boldsymbol{f}_i, \tag{14}$$

**Drag Magnitude:**
$$\boldsymbol{f} = \sum_{i=1}^{k} \frac{|d_i|}{\sum_{i=1}^{k} |d_i|} \boldsymbol{f}_i, \tag{15}$$

where $\boldsymbol{P}_i$ is the pixel-wise distance map to handle point $h_i$, and $|d_i|$ denotes the drag magnitude of $h_i$.

### C.2  VISUAL RESULTS OF MODULATION FACTORS

Moreover, we present the visual results of different combinations of $\alpha$ and $\beta$ in Fig. 22. It is evident that GeoDrag well aligns with user-specified drag points, demonstrating its robustness across parameter variations. We observe that increasing $\alpha$ enhances the influence of geometric guidance during editing. As illustrated in the fourth row of Fig. 22, when $\alpha = 2$, the result exhibits a more plausible 3D transformation: the mushroom cap appears lifted along a realistic vertical trajectory, consistent with a bottom-up viewpoint. This demonstrates the role of geometric priors in maintaining a consistent 3D perspective during editing.

### C.3  QUANTITATIVE RESULTS OF DIFFERENT $\gamma$

The quantitative results of different perceptual region $\gamma$ are presented in Fig. 12. As $\gamma$ increases, MD consistently decreases, while IF exhibits a non-monotonic trend—first increasing, then decreasing. Notably, $\gamma = 1$ offers a favorable balance between editing accuracy and image fidelity, and is thus selected as the default setting in our experiments.

### C.4  ABLATION STUDY ON INVERSION STEPS

We experiment with $t \in \{4, 7, 10, 13, 20, 35, 50\}$ to examine the number of inversion steps in diffusion inversion. The qualitative and quantitative results are given in Fig. 14 and Fig. 13, respectively. We observe that GeoDrag achieves more precise and faithful edits when the number of inversion steps $t \leq 10$. In contrast, when $t > 10$, the editing quality degrades. For a fair comparison with FastDrag (Zhao et al., 2024), and to balance editing performance with fidelity to the original image, we adopt 10 as the default setting for inversion steps throughout all experiments in the paper.

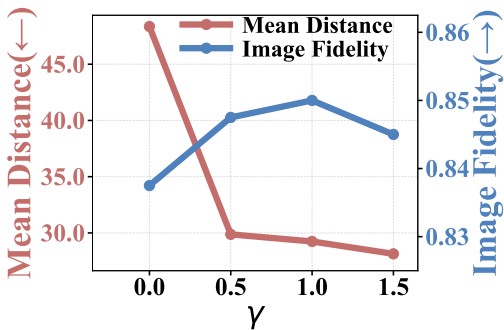

Figure 12: Ablation study on $\gamma$ in terms of quantitative metrics.

Figure 13: Quantitative evaluation of the effect of varying inversion steps in diffusion inversion.

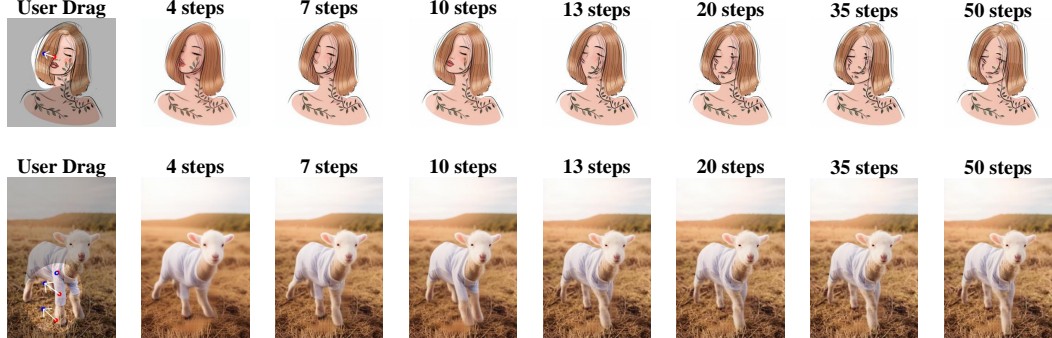

Figure 14: Ablation study on the number of inversion steps

# D  MORE VISUALIZATION COMPARISON

We present additional qualitative comparisons between our method and other state-of-the-art interactive point-based editing methods. As shown in Fig. 23, these results further demonstrate the advantages of GeoDrag across various scenarios, including rotation (e.g., dog, car), scale manipulation (e.g., the avocado and burger), stretching (e.g., the bench), and geometry-consistent movement (e.g., the mailbox and car). Compared to other methods, GeoDrag better preserves object structure while producing edits that better conform to the user input. In challenging cases involving perspective shifts (e.g., the last row), GeoDrag generates geometry-consistent results that maintain alignment with the user-specified drag. These results demonstrate GeoDrag's ability to preserve visual coherence under complex editing operations.

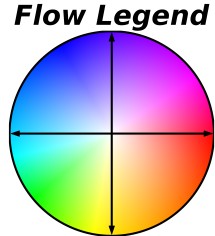

(a) Color wheel used for displacement field visualization.

(b) Color bar used to indicate depth or distance maps.

Figure 15: Visualization legends.

Table 2: Quantitative comparison of GeoDrag with and without LoRA finetuning.

| Method | MD $\downarrow$ | DAI$_1 \downarrow$ | DAI$_{10} \downarrow$ | DAI$_{20} \downarrow$ | IF $\uparrow$ |
|--------|------|--------|----------|----------|------|
| **w/o LoRA** | 29.24 | 0.128 | 0.120 | 0.111 | 0.848 |
| **w/ LoRA** | 30.91 | 0.186 | 0.163 | 0.146 | 0.851 |

## E    EFFECT OF LORA FINETUNING ON GEODRAG

To further investigate the generalization ability of GeoDrag, we evaluate GeoDrag under two configurations: with and without LoRA (Hu et al., 2022) finetuning. Both versions share the same backbone and inference hyperparameters; the only difference is whether LoRA finetuning is applied. As shown in visual examples (see Fig. 24), GeoDrag consistently produces high-quality, geometry-consistent edits even without any finetuning. Nevertheless, LoRA finetuning can enhance local detail and fidelity in some cases (e.g., the fourth row of Fig. 24).

The quantitative results are reported in Table 2. The model without LoRA achieves lower MD and DAI, indicating better alignment with editing guidance. Meanwhile, LoRA finetuning improves visual similarity between the original and edited images (higher IF).

## F    SCALABILITY WITH THE NUMBER OF DRAG POINTS

To evaluate scalability as the number of drag points increases, we conduct a dedicated experiment: 10 representative images are selected and applied 1–8 drag points, yielding a total of 80 edited samples. We measured the average per-image editing time for each point count. As reported in Table 3, the number of drag points has minimal impact on editing time, supporting the practicality of GeoDrag for interactive workflows.

Table 3: Average editing time (in seconds) for different numbers of drag points.

| Number of points | 1 | 2 | 3 | 4 | 5 | 6 | 7 | 8 |
|------------------|---|---|---|---|---|---|---|---|
| **Time (seconds)** | $12.71s$ | $11.63s$ | $11.74s$ | $11.23s$ | $12.27s$ | $12.86s$ | $12.03s$ | $12.93s$ |

## G    EFFECT OF MASKS

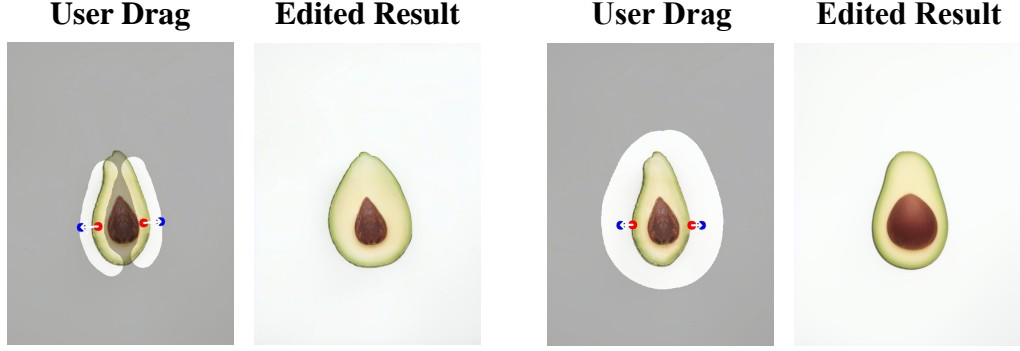

(a) Full-object mask (global edit).                (b) Edge-only mask (localized boundary edit).

Figure 16: Effect of mask selection on editing.

Mask selection is crucial for controlling the scope and locality of edits. Fig. 16 illustrates the impact of different masks when edit a same object. Applying a full-object mask to the avocado enlarges

Table 4: Evaluation of GeoDrag under noisy depth maps

| Noisy Level | MD $\downarrow$ | DAI$_1$ $\downarrow$ | DAI$_{10}$ $\downarrow$ | DAI$_{20}$ $\downarrow$ | IF $\uparrow$ |
|---|---|---|---|---|---|
| **Baseline** | **29.24** | **0.128** | **0.120** | **0.111** | **0.847** |
| $\sigma = 0.01$ | +0.66 | +0.00 | +0.00 | +0.00 | +0.00 |
| $\sigma = 0.05$ | +1.23 | +0.002 | +0.002 | +0.001 | +0.00 |
| $\sigma = 0.1$ | +2.85 | +0.004 | +0.003 | +0.003 | -0.002 |
| $\sigma = 0.5$ | +9.81 | +0.007 | +0.004 | +0.004 | -0.006 |

the entire object (Fig. 16(a)) and the shape of the seed is changed. In contrast, masking only the avocado's edges confines the effect to the boundary (Fig. 16(b)), preserving the shape of seed.

## H VISUALIZATION OF DISPLACEMENT FIELDS

To provide a better understanding of how user-specified drags are propagated, we visualize the 2D displacement fields (see Fig. 1 and Fig. 3). These displacement fields are color-coded using a consistent scheme to indicate direction and magnitude, as shown in Fig. 15(a). Hue represents the direction of displacement, with each color corresponding to a specific motion orientation (e.g., rightward in red, upward in cyan). Brighter and more saturated regions correspond to larger displacement magnitudes.

In addition, Fig. 15(b) shows the legend used for visualizing depth or distance maps. Warmer colors indicate closer regions and cooler colors denote farther ones. This legend is used in visualizations such as depth and distance maps in Fig. 1.

## I DETAILS OF USER STUDY

Here, we provide additional details about the user study, including its design and aggregated ranking-based evaluation results. Fig. 25 shows the selected input images, the corresponding editing results from different methods, and the user-assigned rankings for each result. For each test case, users were asked to rank the edited results from three different methods. The ranking is based on how well each result aligns with the intended dragging operation while preserving the original visual identity. A lower rank indicates better quality. Specifically, rank 1 denotes the best edit—i.e., the one that best aligns with the drag intention and maintains high image fidelity—while rank 3 corresponds to the least satisfactory result. Note that the anonymous labels **(a)**, **(b)**, and **(c)** correspond to FastDrag (Zhao et al., 2024), FreeDrag (Ling et al., 2024), and GeoDrag, respectively.

## J ROBUSTNESS TO DEPTH MAP

Fig. 17 presents a qualitative evaluation of our method under challenging depth-prediction conditions, including transport-specular regions (first row), textureless regions (second row), and a strong perspective view (third row). For each scenario, we compare two variants: geometry-only and the combined formulation employed in GeoDrag. As shown in the figure, the underlying depth maps in these scenes exhibit distinct failure modes: (1) due to the transparent glass surface and strong specular reflections, the depth estimator can only separate foreground and background coarsely, producing a flat and overly simplified depth map. When relying solely on depth cues, this leads to blurred deformation along object boundaries; (2) the edited object lies almost flush against a uniformly colored wall, resulting in an extremely weak depth gradient. Geometry-only guidance, therefore, becomes unreliable and introduces distortions; (3) the strong perspective foreshortening causes large depth discontinuities and makes it difficult for the estimator to recover the true geometry of the target object (the train).

In all these scenarios, the failure of the depth map to provide reliable geometric cues renders geometry-only guidance insufficient for stable deformation. However, the incorporation of Spatial Plane Modulation effectively compensates for these deficiencies by supplying a depth-independent,

| User Drag | Depth Map | w/o Pane | w Pane |
|---|---|---|---|

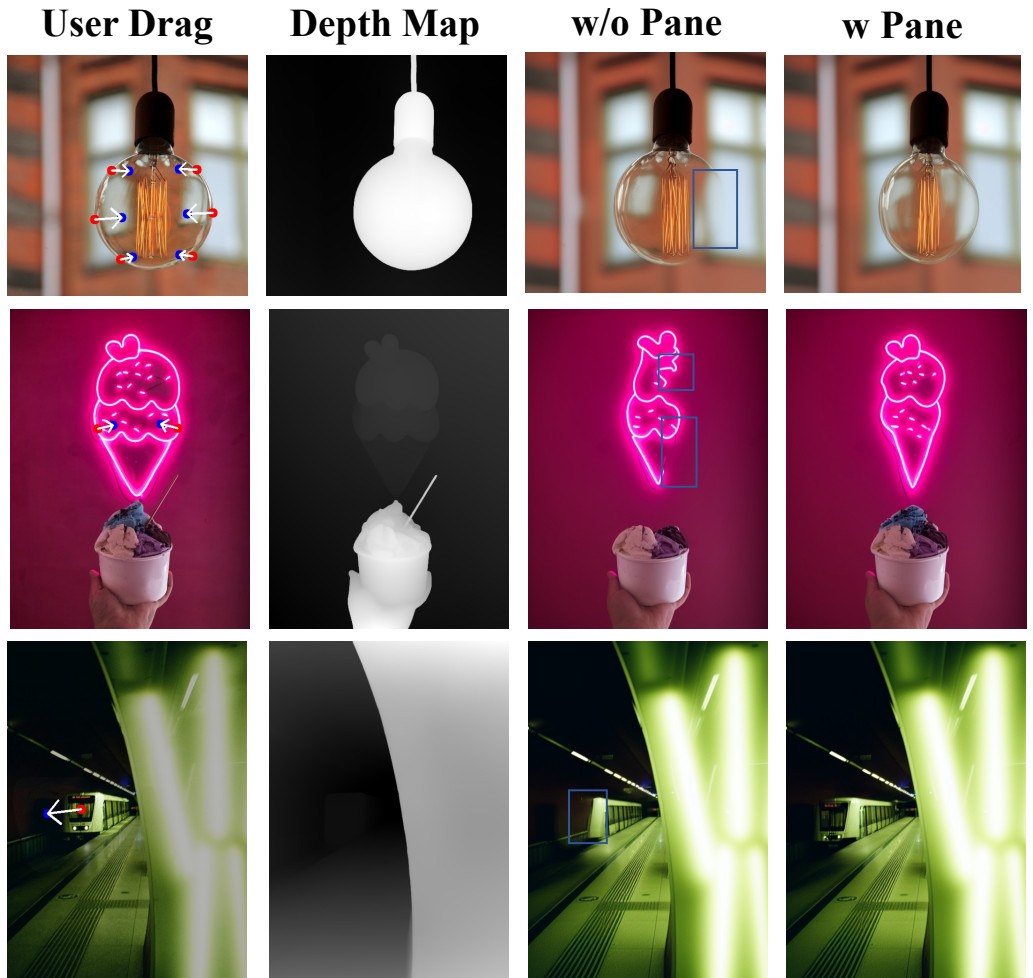

Figure 17: Evaluation in specular, textureless, and strong-perspective scenes using geometry-only, and combined displacement.

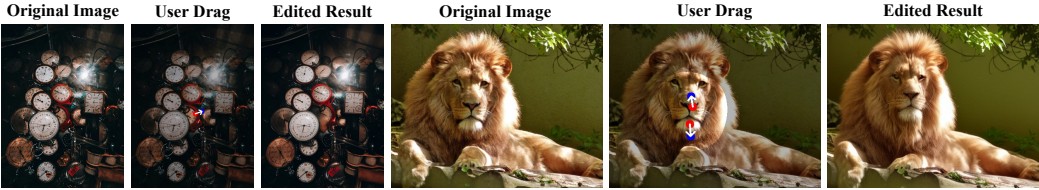

Figure 18: Failure cases of drag editing when generating previously unseen content. Left: overlapping objects cause ambiguous interpolation and distorted regions. Right: large unseen areas lack valid latent correspondence, leading to unrealistic synthesis.

spatially smooth prior that preserves coherent structure even when depth information is severely degraded. This design enables GeoDrag to maintain stable and plausible deformations.

Moreover, we conducted an additional perturbation study to evaluate the robustness of GeoDrag under noisy monocular depth predictions. We injected controlled Gaussian noise into the predicted depth maps, scaling the variance according to the depth range of each image:

$$\tilde{D}(x,y) = D(x,y) + \epsilon(x,y), \qquad \epsilon(x,y) \sim \mathcal{N}\left(0, \, (\sigma \, \Delta D)^2\right), \tag{16}$$

where $D(x,y)$ is the original depth, $\Delta D = D_{\max} - D_{\min}$ denotes the depth range, and $\sigma$ is the noise factor. This formulation ensures that the injected perturbation is proportional to the scene's

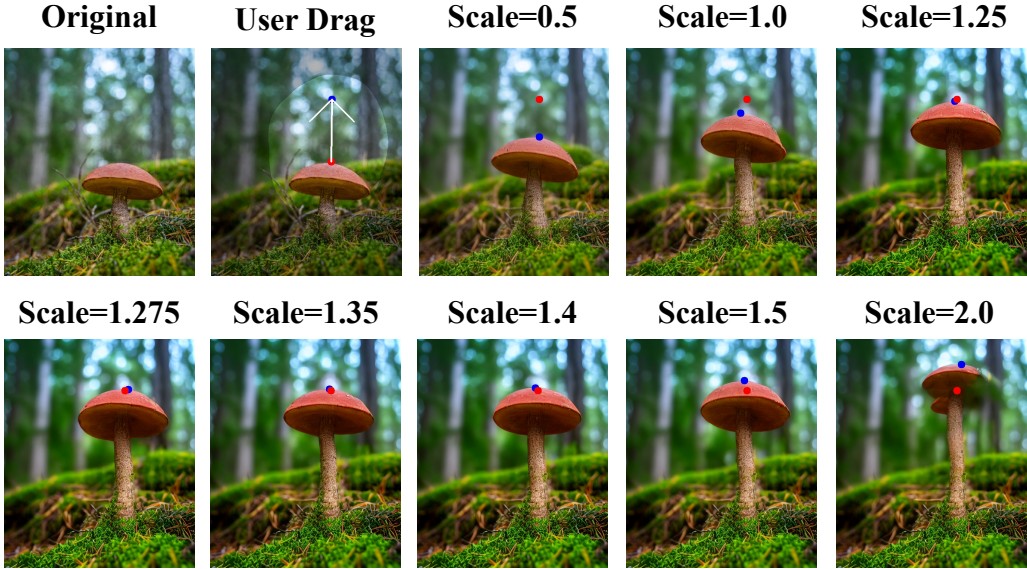

Figure 19: Effect of the scaling factor on handle–target alignment. Increasing the scale improves point alignment, while overly large scales lead to structural distortion.

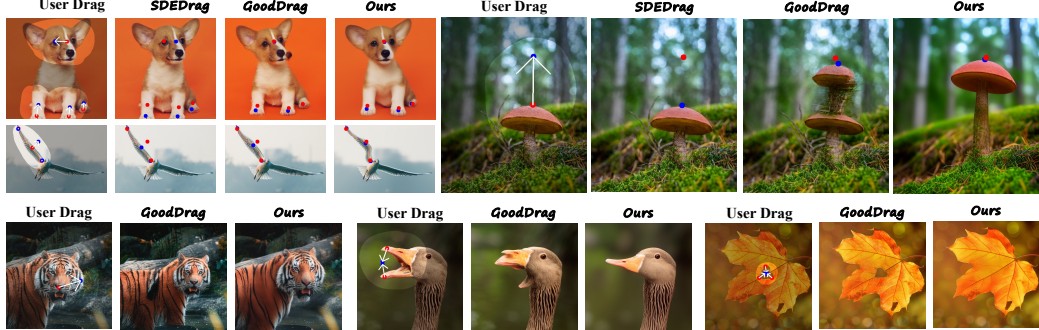

Figure 20: Qualitative comparison with SDEDrag (Nie et al., 2024) and GoodDrag (Zhang et al., 2025).

depth variation. As reported in Table 4, GeoDrag exhibits strong robustness to depth prediction noise. Under mild and moderate perturbations (e.g., $\sigma \leq 0.1$), the performance remains largely stable across all evaluation metrics, indicating that the model is not overly sensitive to small depth fluctuations. Even when the noise magnitude increases to $\sigma = 0.5$, the performance degradation only becomes noticeable at this extreme level, suggesting that the method is capable of maintaining reliable tracking quality under realistic sensor and prediction uncertainties. Overall, these results demonstrate that GeoDrag preserves its effectiveness in challenging noisy-depth scenarios and does not rely heavily on precise depth estimates to function correctly.

## K    FAILURE CASES

We present two representative failure cases in Fig. 18 to clarify the limitations of GeoDrag. Both examples require generating previously unseen content rather than simply deforming the visible regions. The example on the left additionally involves a complex scene with multiple overlapping objects. Although the editing direction is partially correct, the newly exposed regions exhibit distorted and blurry structures. This occurs because our interpolation strategy reconstructs unseen content by sampling nearby latent features; when multiple objects heavily overlap, the mixed semantic context

Table 5: Averaged per-stage execution time of GeoDrag over the full benchmark (in seconds).

| Stage | DDIM inversion | DDIM sampling | Interpolation | Depth prediction |
|---|---|---|---|---|
| Time (s) | 1.2721 | 1.2416 | 0.2247 | 0.0995 |
| Stage | Spatial plane modulation | Latent relocation | Partition | Geometry-aware field modeling |
| Time (s) | 0.0109 | 0.2714 | 0.0006 | 0.0003 |

Table 6: Comparison of dynamic (original) and static (optimized) interpolation in GeoDrag.

| Model | MD $\downarrow$ | DAI$_1 \downarrow$ | DAI$_{10} \downarrow$ | DAI$_{20} \downarrow$ | IF $\uparrow$ | Interpolation Time (s) | Latent Relocation Time (s) |
|---|---|---|---|---|---|---|---|
| GeoDrag (original) | 29.24 | **0.128** | **0.120** | 0.111 | 0.847 | 0.2247 | 0.2714 |
| GeoDrag (optimized) | **27.84** | 0.133 | 0.121 | 0.111 | 0.847 | **0.0014** | **0.0009** |

introduces feature ambiguity and leads to unstable reconstruction. The example on the right exposes a large area that is completely invisible in the original image. In this case, the missing content has no reliable correspondence within the existing latent featureslacks reliable correspondence within the existing latent features, so interpolation-based completion cannot provide meaningful structural cues, resulting inpainting-based latent completion module may alleviate this issue and is a promising direction for future work.

## L    PER-STAGE RUNTIME ANALYSIS AND IMPROVEMENTS

To better characterize the efficiency of the proposed GeoDrag pipeline, we provide a full-stack runtime breakdown across all modules, revealing where the computational budget is primarily spent and how to further optimized. The pipeline can be decomposed into eight major stages: DDIM inversion, DDIM sampling, partition, geometry-aware field modeling, spatial plane modulation, interpolation, latent relocation, and depth prediction. The averaged per-stage execution times (in seconds) over the *Other Object* subset of DRAGBENCH are reported in Table 5. The results show that the diffusion-based components (inversion and sampling) dominate the total latency, whereas the remaining geometric modules contribute negligibly to the runtime. Within the non-diffusion stages, interpolation and latent relocation are most expensive component.

Therefore, we focus our efficiency improvements specifically on these two modules: (1) we observed that latent relocation can be fully vectorized, allowing us to remove unnecessary per-point loops; and (2) we replace the dynamic BNNI update with a static four-neighbor interpolation rule, enabling parallel computation and substantially faster execution. The quantitative comparison is reported in Table 6. Surprisingly, this simplified interpolation not only reduces runtime but also improves performance. We believe this is because removing the iterative BNNI updates prevents error accumulation, while the static strategy still retains the essential semantic structure in the latent space.

## M    EFFECT OF THE SCALING FACTOR ON HANDLE–TARGET ALIGNMENT

During the drag operation, the displacement field is computed in the diffusion latent space rather than directly on image pixels space. When converting drag points from the pixel domain (e.g., $512 \times 512$ pixels) to the latent domain (e.g., $64 \times 64$ features), the coordinates must be downscaled accordingly. This conversion inevitably reduces the effective displacement magnitude in the latent space, leading to a small residual gap between the handle and target points after editing.

To compensate for this attenuation, we apply a scaling factor to rescale the displacement field after converting coordinates to the latent space. This restores the motion magnitude lost during downsampling and improves handle–target alignment.

As shown in Fig. 19, the scaling factor significantly influences alignment and deformation quality. A moderate value enhances correspondence between handle and target points, while an overly large

one leads to overcorrection and geometric distortion. In practice, a scale of 1.2–1.3 offers a good balance between point-level accuracy and global consistency.

## N    ADDITIONAL VISUAL COMPARISONS WITH SDEDRAG AND GOODDRAG

We provide additional qualitative comparisons with SDEDrag (Nie et al., 2024) and Good-Drag (Zhang et al., 2025) in Fig. 20. The results show that our method achieves comparable or even better editing results. Compared with GoodDrag (Zhang et al., 2025), GeoDrag does not rely on LoRA fine-tuning and multi-step optimization, making it significantly more efficient.

## DECLARATION OF LLM USAGE

We used a large language model (GPT-5) solely for grammar and language refinement. All research ideas, analyses, and conclusions are our own.

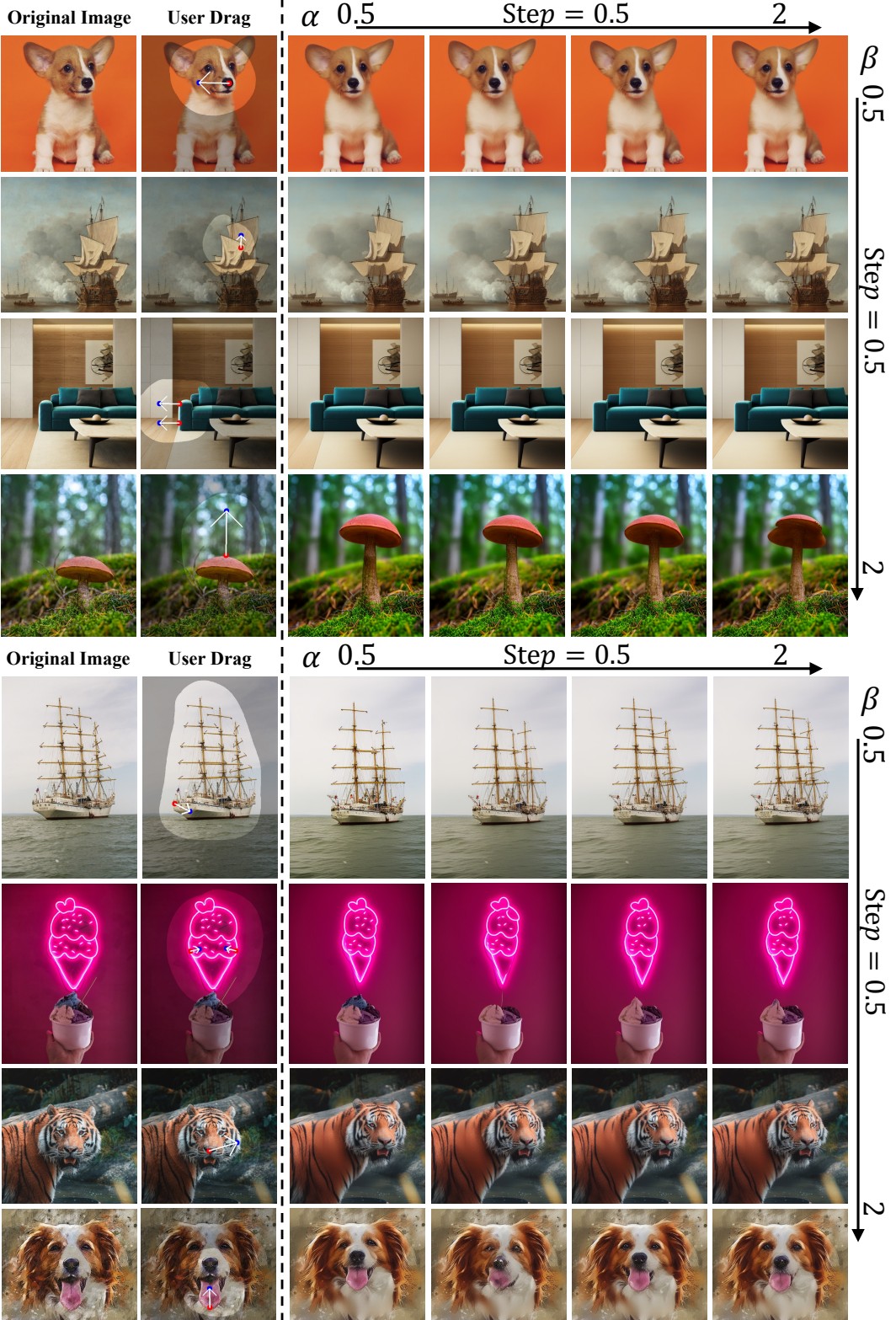

Figure 21: Visual Results of Hyperparameter Analysis.

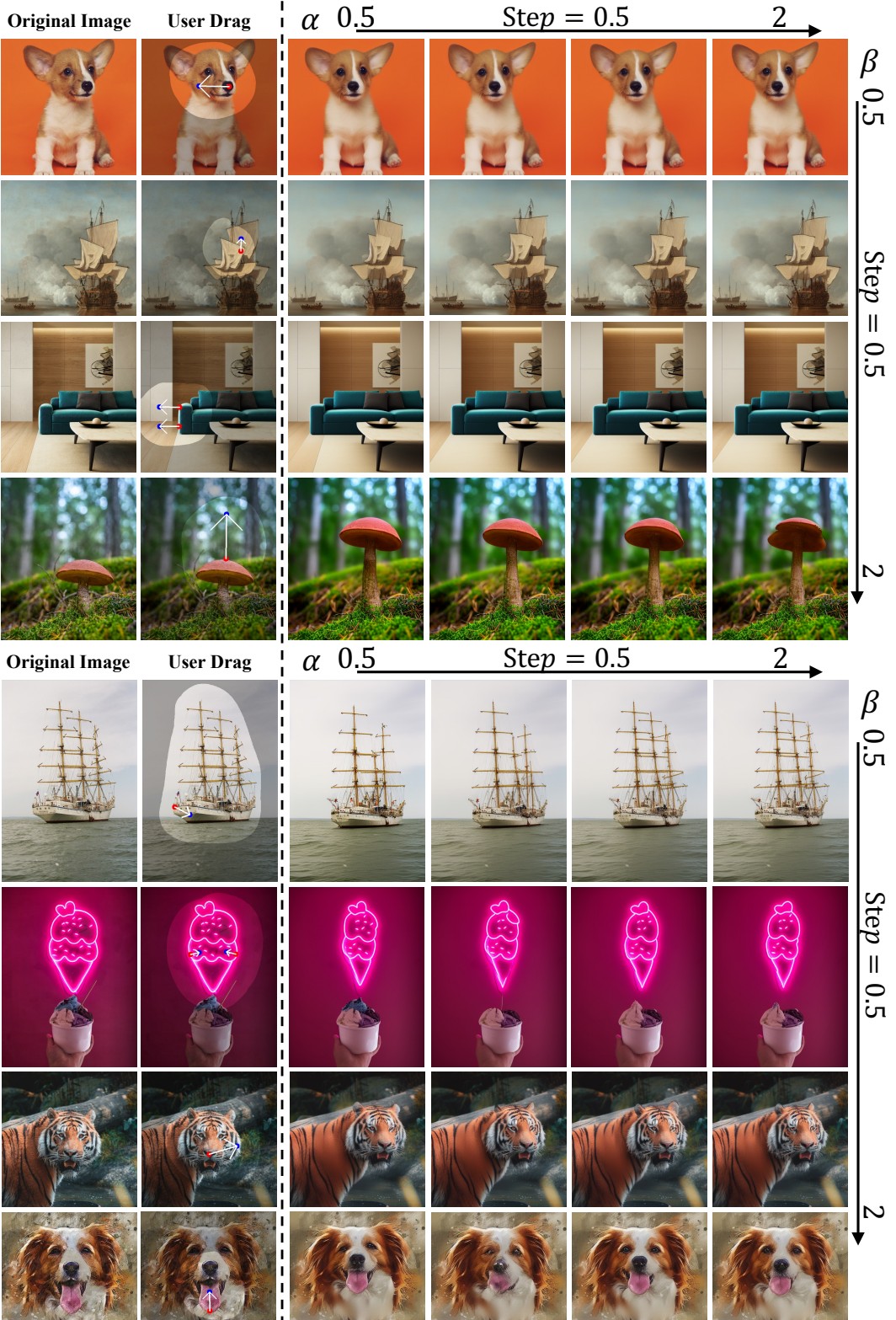

Figure 22: Visual Results of Hyperparameter Analysis.

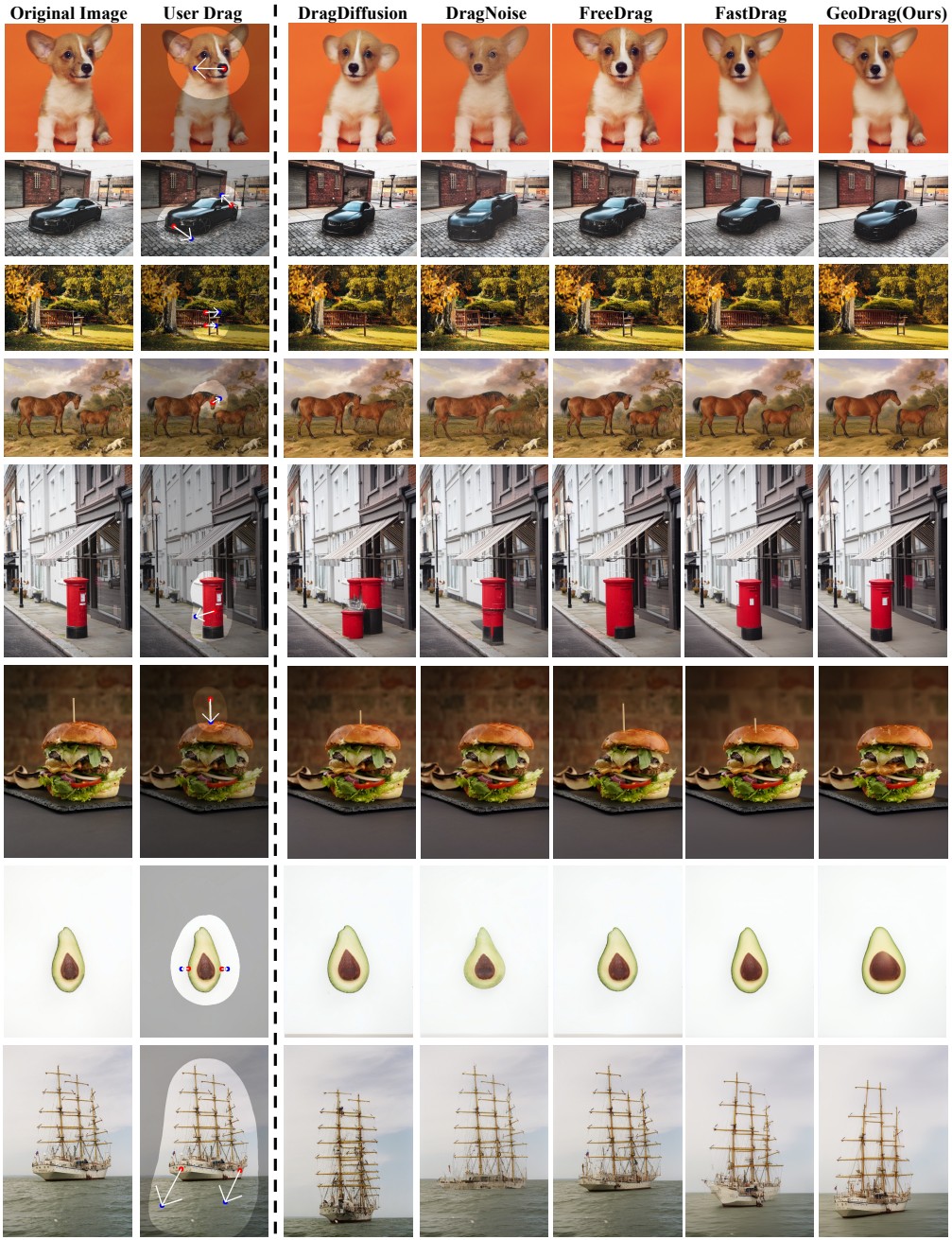

Figure 23: More qualitative comparisons with state-of-the-art interactive point-based methods.

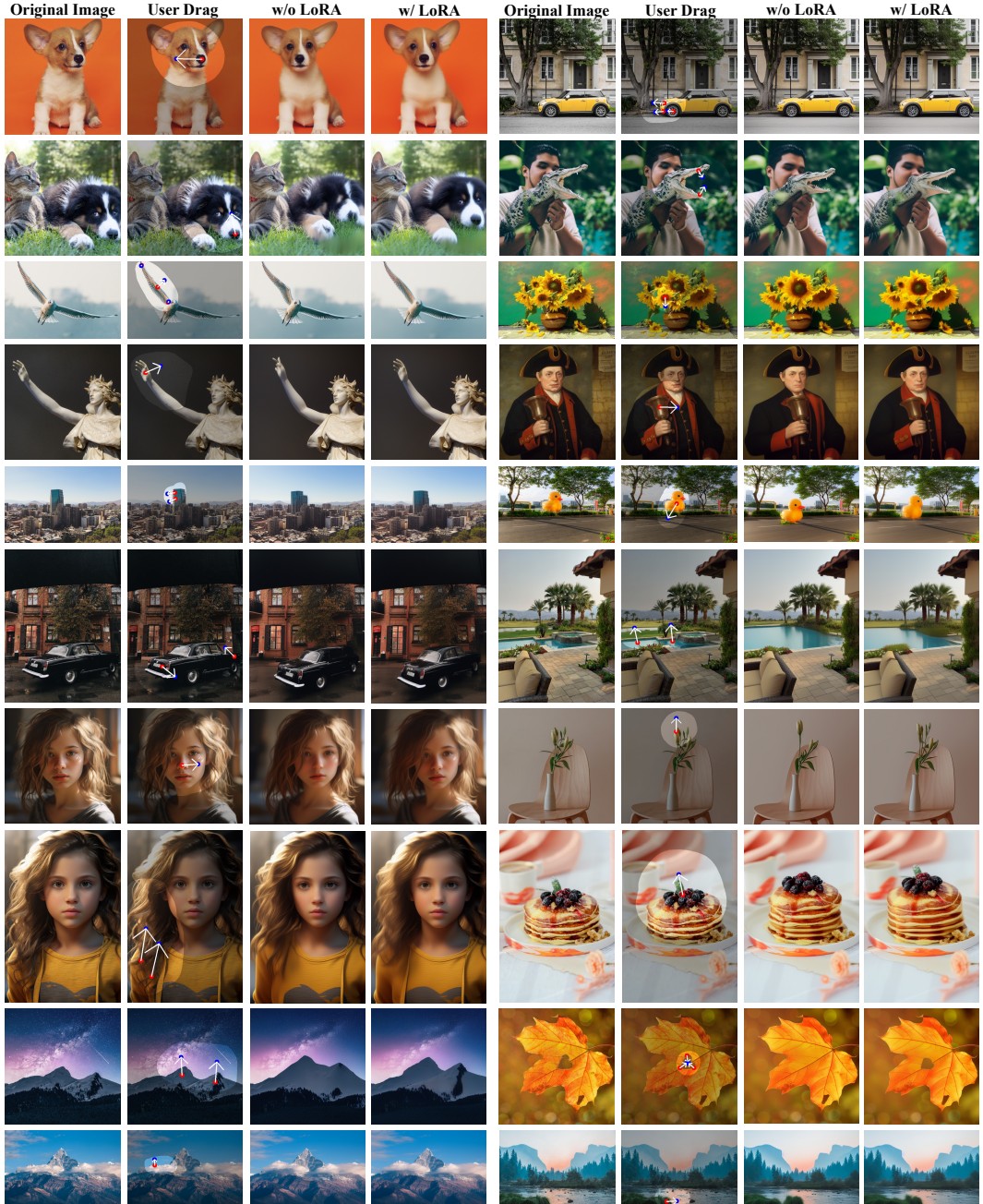

Figure 24: Visual comparison of GeoDrag with and without LoRA (Hu et al., 2022) finetuning. **w/ LoRA** and **w/o LoRA** denote our GeoDrag with and without finetuning, respectively.

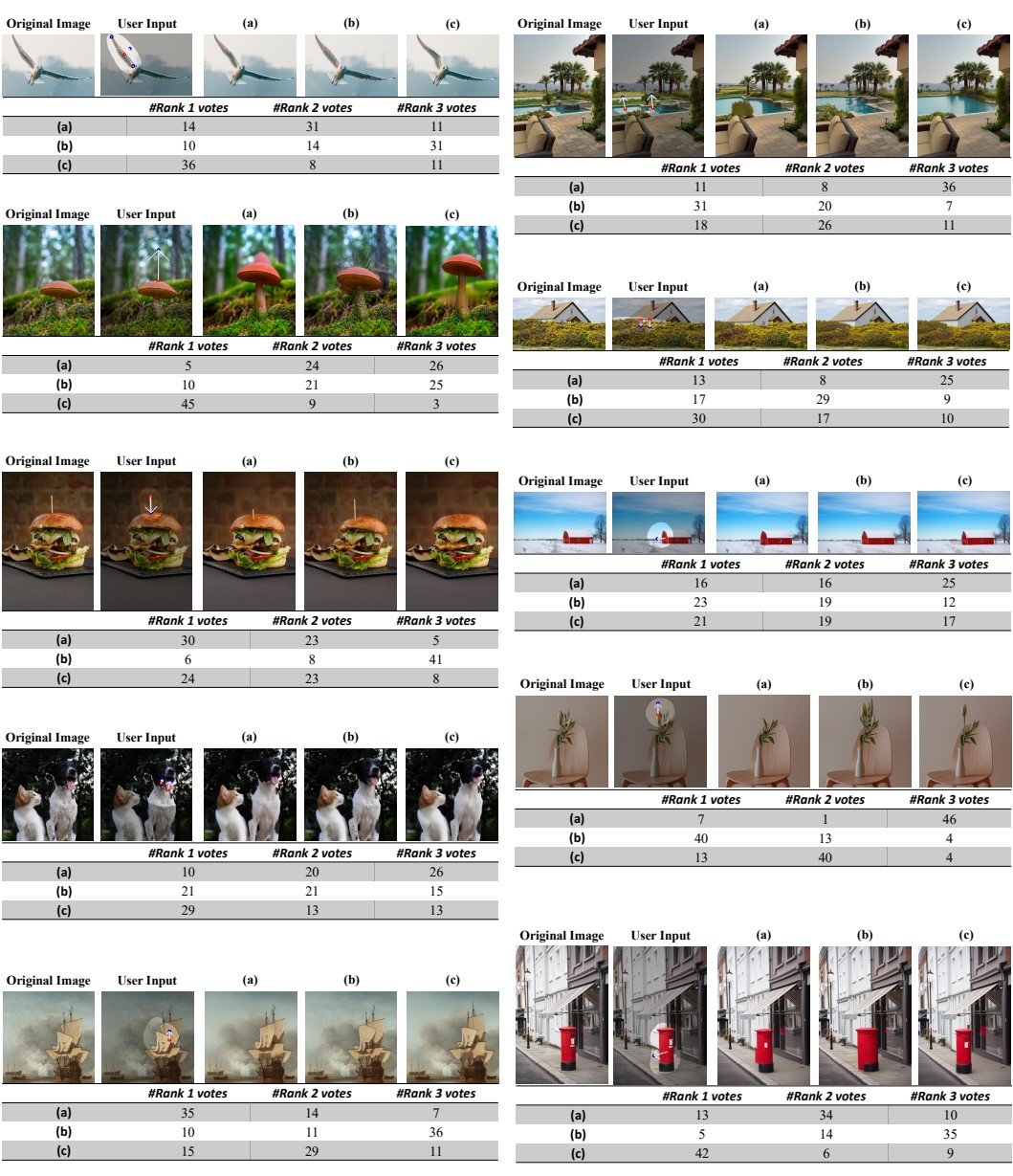

Figure 25: Examples and detailed user rankings for the user study. For each case, we show the original image, the user input, and editing results from three anonymous methods labeled as **(a)**, **(b)**, and **(c)**. The number of Rank-1, Rank-2, and Rank-3 votes collected from all participants is reported. A lower rank indicates better alignment with the intended manipulation and higher visual fidelity.

