# OpenReview forum: "Dragging with Geometry: From Pixels to Geometry-Guided Image Editing"
_ICLR.cc/2026/Conference — ICLR 2026 Poster_

### Official Review · Reviewer_m8ds · 2025-10-25

**Soundness:** 3
**Presentation:** 3
**Contribution:** 2
**Rating:** 4
**Confidence:** 2

**Summary:**

GeoDrag is a geometry-aware drag-based image editing framework combining 3D structural cues and 2D spatial priors. It predicts displacement fields in diffusion latent space, incorporating depth-based modulation and spatial plane adjustment, along with a conflict-free partitioning scheme for multi-point editing.

**Strengths:**

1. Addresses a concrete weakness: Most drag methods ignore geometry, causing distortions. GeoDrag effectively solves this.

2. Principled integration: The combination of 3D depth modulation and 2D plane adjustment is physically intuitive and well-designed.

3. Practical contribution: One-step editing drastically improves usability and speed over iterative gradient-based methods.

4. Strong empirical results: Quantitative improvements (1.4× DAI, 1.1× MD) and excellent qualitative visuals demonstrate real impact.

5. Multi-point conflict resolution: - The conflict-free partitioning module is a simple yet powerful idea that improves robustness.
6. Extensive evaluation: Diverse examples (faces, landscapes, multi-object scenes) show generality.

**Weaknesses:**

1.	Depth dependency: Accuracy hinges on reliable depth maps; the paper assumes ideal depth input without addressing noise robustness.
2.	Limited theoretical grounding: The interaction between geometry and diffusion latent space lacks mathematical explanation.
3.	Ablation insufficiency:The individual contribution of geometry-aware vs. plane-aware modules is unclear.
4..	Efficiency discussion shallow: Though claimed efficient, actual GPU/latency data is minimal.
5.	Applicability scope: Focused on still images; no experiments on video or non-diffusion backbones.
6..	Minor clarity issues: Some figures are over-compressed, making displacement fields hard to interpret.

**Questions:**

1.	How sensitive is the method to depth map errors?
2.	Can GeoDrag operate with estimated depth from monocular models?
3.	Is the diffusion model re-trained or used as a frozen backbone?

---

> ### Author Response · Authors · 2025-11-22
> **Response to Reviewer m8ds (Part 1)**
>
> Thank you for the insightful and valuable comments! Below we present our point-by-point responses, which we hope will address your concerns.
> 1. **Depth dependency.** We thank you for raising this concern. Depth estimation can be unreliable in a few real-world scenarios, such as specular highlights, textureless surfaces, or strong perspective views. Monocular depth maps can indeed degenerate into overly smooth planes (e.g., walls) or break around specular regions. In such cases, relying solely on geometry-aware displacement would be unstable. Therefore, we incorporate **Spatial Plane Modulation**, which computes the displacement field purely from pixel distances and does not rely on depth gradients. This ensures that the method offers stable deformations whenever depth becomes unreliable.
>     - To further validate this behavior, we **added Appendix J (Fig. 17)** in the revised manuscript, where we analyze these challenging scenarios and examine whether spatial plane modulation can mitigate depth-prediction noise. We include cases with specular reflections, textureless regions, and strong perspective view—cases where monocular depth often fails—show that the modulation maintains stable and coherent deformations even under severe depth inaccuracies.
>     - In addition, we conduct **a newly added depth-noise perturbation study presented in Appendix J** to further assess the robustness of GeoDrag. We simulate noisy depth estimation by injecting Gaussian noise proportional to each image’s depth range: $$\tilde{D}(x,y)=D(x,y)+\epsilon(x,y),\quad\epsilon(x,y)\sim\mathcal{N}\bigl(0,;(\sigma,\Delta D)^2\bigr),$$ where $D(x,y)$ is the predicted depth, $\Delta D=D_{\max}-D_{\min}$ denotes the scene depth range, and $\sigma$ controls the noise level. The quantitative outcomes are summarized below and **Appendix J (Table 4)**. While increasing noise gradually reduces accuracy, GeoDrag remains stable under small and moderate perturbations. Only when the noise becomes extremely large (e.g., $\sigma=0.5$) does the performance significantly drop, confirming that GeoDrag can still perform reliably even when depth predictions are imperfect.
>
>         |$\sigma$|MD$\downarrow$|DAI$_1$$\downarrow$|DAI$_{10}$$\downarrow$|DAI$_{20}$$\downarrow$|IF$\uparrow$|
>         |-|-|-|-|-|-|
>         |**Baseline**|**29.24**|**0.128**|**0.120**|**0.111**|**0.847**|
>         |$0.01$|+0.66|+0.000|+0.000|+0.000|+0.00|
>         |$0.05$|+1.23|+0.002|+0.002|+0.001|+0.00|
>         |$0.10$|+2.85|+0.004|+0.003|+0.003|-0.002|
>         |$0.50$|+9.81|+0.007|+0.004|+0.004|-0.006|
>
> 2. **Theoretical grounding.** Thank you for pointing this out. We clarify the theoretical basis of the interaction between geometry and the diffusion latent space from the following two perspectives.
>    1. **Mathematical grounding from geometry.** Although the editing instruction is provided directly in the 2D image plane, the physical transformation it implies inherently reflects the underlying 3D structure of the depicted object. Under a perspective projection, points closer to the camera undergo larger apparent motion in the image plane than points farther away (Wang et al., "Zolly: Zoom Focal Length Correctly for Perspective-Distorted Human Mesh Reconstruction," ICCV 2023). For instance, when rotating a human head, points closer to the camera undergo greater displacement in the projected image than points farther away; this effect originates from the perspective projection model. Based on this observation, we derive the geometry-aware displacement field in a principled manner by explicitly projecting 3D drag displacements onto the 2D image plane. Concretely, starting from the perspective projection model, we analyze how a 3D translation $(\delta x, \delta y, \delta z)$ on the object surface induces its corresponding pixel-plane motion $(\delta u, \delta v)$. This derivation leads to a depth-dependent formulation, where pixels closer to the camera exhibit larger projected motion and farther pixels exhibit smaller motion. Leveraging this depth-consistent projection relationship, we further derive a closed-form scaling rule (Eq. 3-7) that converts a 2D drag instruction and a depth map into a geometry-aware displacement field.
>    2. **How the geometry-aware field interacts with diffusion latent space.** Our method builds on a well-established property of diffusion models: their latent representations preserve strong spatial and semantic correspondence with the pixel space. Thus, modifying a localized latent region typically yields a semantically aligned change in the corresponding pixel area. **This property is also the basis of prior drag-based editing methods (DragDiffusion (CVPR 2024), EasyDrag (CVPR 2024), FastDrag (NIPS 2024), FreeDrag (CVPR 2024), NoiseDrag (CVPR 2024), GoodDrag (ICLR 2025))**, which depend on latent–pixel correspondence for spatial control. Our method follows this same principle, but introduces geometric cues to more effectively guide the latent modification.

---

> ### Author Response · Authors · 2025-11-22
> **Response to Reviewer m8ds (Part 2)**
>
> 3. **About ablation study.** We would like to clarify that the individual contributions of the geometry-aware and plane-aware displacement fields had already been evaluated in the original submission. The qualitative and quantitative ablation results are provided in **Section 4.3 (Fig. 8) and Appendix C.1 (Fig. 10(a))**, respectively. The results consistently show that the combination of both achieves the best performance across visual fidelity and editing accuracy, indicating that the two modules are complementary.
> 4. **Efficiency discussion.** We appreciate your interest in the efficiency aspect. For clarification, GPU memory and latency measurements are already part of the original submission. Table 1 reports memory consumption and per-point editing time, and Table 3 in the Appendix shows how the editing time scales with the number of control points. These results demonstrate that our model is highly efficient, the per-point latency is low, and the overall runtime remains nearly constant even when the number of editing points increases, confirming that our method does not incur additional computational overhead with more edit instructions.
>
>    In addition, to further address your concerns, we have included a new experiment that provides a per-stage efficiency analysis. The detailed timing breakdown is reported in **Appendix L (Table 5)** and reproduced below. This profiling includes the main components of our pipeline: depth prediction, spatial partitioning, computation of geometry-aware and plane-aware fields, latent relocation, interpolation, as well as DDIM inversion and sampling. The analysis is conducted on the Other Object subset of DRAGBENCH (35 samples).
>
>     | Stage | DDIM inversion | DDIM sampling | Interpolation | Depth prediction | Plane-aware field computation | Latent relocation | Partition | Geometry-aware field computing |
>     |-------|----------------|---------------|---------------|------------------|----------------|---------------|---------------|------------------|
>     | Time (s) | 1.2721 | 1.2416 | 0.2247 | 0.0995 |  0.0109 | 0.2714 | 0.0006 | 0.0003 |
>
>    The results show that, besides DDIM inversion and sampling, the most time-consuming stages are latent relocation and interpolation. Therefore, we accelerate GeoDrag by improvementing on these two modules.
>     - Latent relocation. In the original implementation, latent relocation iteratively updates each point in a loop, resulting in sequential dependency and limited parallelism. We observe that the relocation formula is fully vectorizable; each pixel can be relocated independently without relying on intermediate updates. By rewriting the implementation to operate on batched tensors rather than per-point loops, the entire relocation process becomes parallelizable, eliminating the sequential bottleneck.
>     - Interpolation. The original BNNI strategy is dynamic, each location depends on updated values of other locations, enforcing a sequential scan over the latent grid. We replace this with a static nearest-four interpolation rule, where each location is interpolated exactly once based on its nearest four spatial neighbors (up/down/left/right). This removes the iterative dependency update and allows interpolation to be computed in a single parallel pass.
>
>     As shown in table below, we achieve notable speedups while even improving the editing quality. We attribute this to the removal of iterative dependency updates that can accumulate error, while the static interpolation still preserves sufficient spatial coherence in the latent domain.
>
>     | Model | MD $\downarrow$ | DAI$_1$$\downarrow$ | DAI$_{10}$ $\downarrow$ | DAI$_{20}$ $\downarrow$ | IF $\uparrow$ |     Interpolation time | Latent relocation time |
>     |-------|----------------|----------------------|-------------------------|-------------------------|----------------|    ---------------------|------------------------|
>     | GeoDrag (dynamic interpolation / original) | 29.24 | **0.128** | **0.120** | 0.111 | 0.847 | 0.2247 | 0.2714 |
>     | GeoDrag (static interpolation / optimized) | **27.84** | 0.133 | 0.121 | 0.111 | 0.847 | **0.0014** | **0.0009** |

---

> ### Author Response · Authors · 2025-11-22
> **Response to Reviewer m8ds (Part 3)**
>
> 5. **Clarification of our scope.** We appreciate your concern regarding the scope of applicability. We would like to clarify the intended scope of our work. Our method is **deliberately scoped for interactive point-based editing of still images**, and is **built upon diffusion-based image generation models**. The focus of this paper is to **improve controllability and performance of interactive image editing within the diffusion paradigm**, rather than proposing a general editing framework applicable to video or non-diffusion architectures.  This design choice reflects the actual structure of the method rather than a limitation of the experimental setting.
>
>     Importantly, image editing and video editing belong to fundamentally different problem paradigms. Video editing requires addressing temporal consistency, motion trajectories, and cross-frame correspondence, which are not part of the problem our method is designed to solve, and solving them would require architectural and algorithmic components beyond the scope of this paper. For this reason, extending our method to videos is not a matter of adding experiments, but would constitute a different research problem.
>
>     For other point-based dragging approaches (e.g., DragDiffusion (CVPR 2024) and FastDrag (NeurIPS 2024)), their designs—like ours—are tightly coupled to the diffusion denoising and sampling process. Methods in this family are not expected to be directly portable to non-diffusion generators, because their core mechanisms (e.g., iterative denoising, noise conditioning, diffusion-specific latent manipulation) assume a diffusion architecture. Therefore, applying our design or those of other diffusion-based dragging methods to non-diffusion backbones would effectively require developing a new method, rather than constituting a straightforward extension or a meaningful ablation of our current framework.
>
>     We fully agree that extending interactive point-based editing to videos or to non-diffusion backbones is an exciting and important future direction. The conceptual insights of our work may inspire such extensions, but they are intentionally left as future work and are beyond the scope of the present paper.
>
> 6. **Figure clarity.** Thanks for pointing this out. We would like to clarify that all figures are rendered at the original pixel resolution and exported to PDF to retain sharpness. Regarding the displacement field visualization, we assume the comment refers to Fig. 3. In that figure, our intention is to illustrate the combined displacement field formed by blending the geometry-aware and plane-aware fields. To keep every component visible within the full framework layout, the displacement field visualization is scaled accordingly. Enlarging the figure further within the paper would compromise its visual layout and exceed the available page limit. To support clearer inspection, a higher-resolution version of the framework figure is provided on our anonymous project page, where the displacement field can be viewed in greater detail. We would be very happy if the reviewer could refer to that resource for a more precise visualization.
> 7. **Monocular depth support.** Our method already operates with monocularly estimated depth. All experiments in the paper use depth maps predicted by the Depth Anything V2 (NeurIPS 2024) model rather than ground-truth or multi-view depth.
> 8.  **Frozen diffusion backbone.** The diffusion model is used as a frozen backbone. Our entire method is training-free and does not require any fine-tuning of the diffusion model.

---

> ### Author Response · Authors · 2025-11-28
>
> Thank you once again for your thoughtful review and valuable comments! With only seven days remaining before the end of the Discussion Phase, we would greatly appreciate your feedback on our revised responses. Please let us know whether our clarifications adequately address your concerns. If any issues remain unresolved or if further explanation is needed, we would be grateful for your additional guidance.
>
> Best regards, ﻿
>
> Authors

---

### Official Review · Reviewer_2Gtw · 2025-10-30

**Soundness:** 3
**Presentation:** 3
**Contribution:** 3
**Rating:** 6
**Confidence:** 4

**Summary:**

This paper presents GeoDrag, a novel method for point-based image editing that addresses the limitations of 2D-only approaches in geometry-intensive scenarios. By incorporating 3D geometric cues (from a predicted depth map) into a unified displacement field, GeoDrag achieves structurally consistent edits like rotations and perspective shifts. The framework combines this geometry-aware guidance with 2D spatial priors for local detail preservation and introduces a conflict-free partitioning strategy for robust multi-point dragging, all within an efficient single-pass process.

**Strengths:**

1.  **Novelty and Rationale:** The core strength is the novel integration of 3D geometry into the 2D drag-editing task. This is a well-motivated and logical advancement for the field, directly addressing a key limitation of prior work and yielding significant improvements in structural preservation.

2.  **Clear and Effective Framework:** The paper effectively decomposes the problem into distinct sub-challenges and proposes clear, corresponding solutions (geometry-aware field, spatial modulation, and conflict-free partitioning). This systematic approach enhances the method's robustness and clarity.

3.  **Efficiency for Interactivity:** By avoiding iterative optimization, the one-step editing process makes GeoDrag a practical and competitive solution for real-world interactive applications.

**Weaknesses:**

1.  **Texture and Identity Preservation:** In some cases, the edited regions exhibit a loss of texture fidelity. For instance, the "mountain" example on the project page shows the dragged peak becoming blurred, suggesting a potential trade-off between geometric deformation and preserving the original surface identity. A discussion on this limitation would be welcome.

2.  **Foreground-Background Separation:** The method can sometimes cause unnatural co-movement of the background with the dragged object. This is visible in the "duck" example, where the surrounding grass is pulled along with the head. This suggests that handling disocclusion and complex object boundaries remains an area for improvement.

3.  **Limitations of the 3D Motion Model:** The current geometric model (Eq. 7) effectively handles transformations parallel to the image plane but is a simplification of true 3D motion. This might limit its ability to perform more complex perspective changes, such as pulling an object "out of" the scene. Exploring a more comprehensive 3D motion model could be a valuable future direction.

**Questions:**

1.  Concurrent work like `Inpaint4Drag` uses inpainting to handle occlusions, which can sometimes cause warping distortions. Have the authors considered combining GeoDrag's superior geometry-aware displacement with a dedicated inpainting model? This synergy could potentially yield more realistic results at object boundaries by better handling disoccluded regions.

---

> ### Author Response · Authors · 2025-11-22
> **Response to Reviewer 2Gtw (Part 1)**
>
> Thank you for the insightful feedback and positive comments! We summarize our responses to each point below. We hope our responses adequately address your concerns.
> 1. **Texture and identity preservation.** The blurry in the project visualization is due to setting $\eta$ in Eq. (2) to 0, which disables texture reasoning during editing. We have **added this case to Appendix C.3 (Fig. 14)**, where using a non-zero $\eta$ produces significantly better texture inference and preserves the mountain appearance. As shown in Fig. 14, when $\eta = 1$ (stochastic), the mountain texture is generated even more faithfully, further supporting our conclusion that introducing controlled stochasticity yields sharper and more faithful refinement.
> 2. **Foreground–background separation.** We concur that when the foreground and background share similar texture or lie in shallow depth differences (e.g., the duck and the surrounding grass), the background may co-move with the foreground during dragging. This is a challenging case for all dragging-based editing frameworks, as disocclusion and boundary complexity cannot be fully addressed through motion reasoning alone. While GeoDrag already improves stability compared to purely 2D approaches, we recognize that foreground–background disentanglement remains an open problem.
>
>     Technically, GeoDrag operates on a single-layer latent representation with a continuous displacement field inside the user-specified mask. When the mask covers both the foreground object and some adjacent background, the displacement filed naturally propagates deformation within this region. As a result, parts of the parts of the background receive non-negligible displacement and appear to move with the object.
>
>     In practice, we find that this issue is most pronounced in challenging cases where (1) the object and background have comparable depths and highly entangled structures, and (2) the editing mask is relatively large. In many other examples with clearer depth separation or tighter masks, GeoDrag maintains good foreground–background separation. We see this as an important direction for future work. A promising extension is to combine GeoDrag with explicit foreground–background modeling, for example by integrating segmentation cues and an inpainting module specialized for disoccluded regions.
>
> 3. **3D motion model.** Thank you for bringing up this valuable point. Our geometry-aware field modeling is indeed based on depth-guided 2.5D projection rather than a full 3D motion representation. As you correctly pointed out, the current formulation mainly models planar displacement fields; however, it leverages 3D geometric cues to construct these fields in a physically meaningful and geometry-consistent manner. This simplification is intentional and motivated by two considerations:
>     1. **Interactive editing efficiency and stability.** Our goal is to support responsive, real-time user interaction. Explicit 3D reconstruction or full 3D motion estimation significantly increases computational overhead and makes the system unsuitable for interactive point-based editing. The proposed 2.5D formulation provides a strong trade-off by improving geometric consistency with no additional optimization or multi-view inference.
>     2. **Task relevance.** In typical drag-based editing scenarios, user intentions are overwhelmingly dominated by in-plane structural deformation and moderate depth-aware displacements rather than camera-viewpoint changes. In this regime, our geometry-aware field modeling already yields substantial improvements in realism and geometry consistency, as evidenced by both quantitative metrics and visual results.
>
>     While "pulling an object out of the scene" is an interesting notion, we would like to clarify that this type of manipulation is inherently challenging for any drag-based editing system. Drag operations are fundamentally designed to deform or relocate objects that exist within the current scene. In contrast, dragging an object outside the observable scene effectively requires synthesizing geometry and appearance that are not present in the image (akin to object removal / new-object generation), which is a significantly more challenging problem than conventional drag-based deformation.
>
>     That said, we fully agree with you that a more comprehensive 3D motion formulation may in principle facilitate drag edits. Modeling full 3D motion could explicitly infer or hallucinate occluded and out-of-plane structures rather than relying on 2.5D projection alone. We consider this a promising future research direction, although achieving it in an interactive, single-image editing setting would require advances in lightweight 3D reasoning and stable completion of unseen geometry.

---

> ### Author Response · Authors · 2025-11-22
> **Response to Reviewer 2Gtw (Part 2)**
>
> 4. **Combine with Inpaint4Drag.** We sincerely thank you for the constructive suggestion and for pointing out the connection to concurrent work such as Inpaint4Drag. We agree that inpainting-based occlusion handling is complementary to our geometry-aware displacement field, and we consider this an exciting direction for extending GeoDrag.
>
>     We are actively exploring how to integrate our geometry-aware field modeling with a dedicated inpainting module. Inpaint4Drag provides valuable inspiration in this regard. Compared with interpolation-based completion, inpainting has a key advantage: it is designed to synthesize content that is not visible in the original image. This bypasses the intrinsic limitation of interpolation, which reconstructs disoccluded regions by sampling surrounding latent features and may therefore produce missing or blurry textures when such features are insufficient or ambiguous. Inpainting, on the other hand, can hallucinate plausible structure and texture for newly exposed regions rather than relying solely on nearby latent features. Inpainting is naturally suitable for filling in these regions, while our geometry-aware field can provide precise spatial guidance and structural consistency for the inpainting module during drag-based editing. We believe such a combination could substantially enhance edit quality in cases involving occlusion and disocclusion.
>
>     To summary, we envision that:
>     - the geometry-aware displacement field can provide more reliable deformation, leading to more consistent edits.
>     - the inpainting module can recover fine-grained textures that interpolation alone fails to preserve.
>
>     We appreciate the reviewer for pointing out this connection. We consider the integration of geometry-aware displacement modeling and inpainting-based latent completion an exciting and valuable direction for future development of GeoDrag.

---

### Official Review · Reviewer_KdHj · 2025-10-31

**Soundness:** 3
**Presentation:** 3
**Contribution:** 2
**Rating:** 6
**Confidence:** 4

**Summary:**

This paper proposes GeoDrag, a geometry-guided drag-based image editing method. Similar to FastDrag, it enables drag-based manipulation without LoRA and operates in a single forward pass. The key idea is to incorporate 3D geometric cues into 2D drag editing through geometry-aware field modeling, where displacement strength is modulated based on the depth map. In addition, the authors introduce spatial plane modulation to complement this, since geometry-aware motion distributes influence uniformly in 3D space, making it less sensitive to local details. The fusion of geometry and plane-aware fields allows the method to maintain both global structure and fine-grained local control (conceptually related to warpage vector modeling in FastDrag). Experiments on DragBench demonstrate that GeoDrag achieves lower Mean Distance (MD) and Dragging Accuracy Index (DAI) than previous state-of-the-art methods, while performing editing faster and without additional training overhead.

**Strengths:**

- GeoDrag achieves the best performance on DragBench, showing the lowest MD and DAI metrics while maintaining competitive image fidelity and efficiency.
- The framework is simple yet effective, enabling drag-based editing in a single forward pass without LoRA tuning, similar to FastDrag but extended with geometry awareness.
- By introducing geometry-aware field modeling and spatial plane modulation, the method is able to integrate 3D geometric cues into 2D editing, improving structure preservation and local controllability.
- The proposed conflict-free partitioning strategy effectively handles multi-point editing, preventing interference between overlapping drags and ensuring stable, coherent results.

**Weaknesses:**

- In Section 3.2, the paper mentions that the geometry-aware displacement field “distributes influence uniformly in 3D space, making it less responsive to subtle and local deformations in 2D image plane.” However, this concept is not intuitive; additional explanation or visual illustration would help readers understand it better.
- The method heavily relies on accurate depth maps. In scenes where the depth estimation is noisy, such as generated images or complex high-frequency scenes with multiple overlapping objects, the geometry-aware field may be incorrectly computed, potentially degrading results compared to purely 2D approaches. Failure case analysis would strengthen the paper.
- Simply projecting 3D depth information into 2D space may not fundamentally resolve geometric ambiguity, since projection inherently loses spatial consistency across varying viewpoints.
- The novelty of the plane-aware field in Section 3.2 feels somewhat limited. Its formulation resembles the displacement weighting in FastDrag, which also applies a distance-based decay for local control. The paper would benefit from a clearer analysis or ablation demonstrating how the proposed plane-aware formulation improves upon or differs from *FastDrag* in practice.
- In Eq. (10), the geometry-aware and plane-aware fields are linearly interpolated with λ = P / (P + γ). This λ depends only on 2D distance and ignores depth variation, thus disconnecting from the true 3D structure. It would be interesting to see whether defining λ as a combined function of both distance and depth difference could yield a more geometrically consistent fusion.
- Missing comparison with GoodDrag (ICLR 2025), a recent baseline for drag-based editing.

**Questions:**

As mentioned above, in Eq. (10), λ is defined as P / (P + γ), which does not incorporate depth. Have the authors considered using a joint distance–depth weighting function to better reflect 3D spatial relationships?

---

> ### Author Response · Authors · 2025-11-22
> **Response to Reviewer KdHj (Part 1)**
>
> Thank you for the insightful and vaulable comments! Below, we provide a point-by-point response and hope our response helps address your concerns. We also look forward to further discussion, which may continue to help refine and resolve the remaining open issues.
>
> 1. Thank you for pointing this out. The geometry-aware field assumes locally smooth 3D motion, and therefore works well when the object undergoes coherent structural deformation. However, relying solely on the geometry-aware field can lead to failure in certain cases. For example, as shown in Fig. 1(b), the leaf contains a sharp depth discontinuity around the mid-rib, which produces an inconsistent displacement field and eventually leads to editing failure. In the **additional Appendix J (Fig. 17)**, we include failure cases of using only the geometry-aware displacement field, such as transport-specular regions (first row), textureless regions (second row), and strong perspective views (third row). The accompanying depth predictions show that the depth map does not provide reliable structural cues in these scenarios, while the desired edits are primarily local (e.g., adjusting leaf textures or modifying a small logo on a wall). Due to this locality and the unreliability of depth in these cases, spatial plane modulation is required to provide stable and locally consistent deformation.
> 2. Thank you for the insightful comment. We agree that the geometry-aware field depends on the quality of the depth map, and failure can occur when the depth estimation is unreliable. We have added analysis of multiple challenging scenarios, as provided in **Appendix J (Fig. 17)**: transport-specular regions (first row), complex textureless areas (second row), and strong perspective views. In these cases, the inaccuracy of the depth map leads to distorted geometry-aware deformation and consequently suboptimal editing results. However, the spatial plane modulation effectively compensates for this issue by providing locally consistent adjustment when depth cues are unreliable. It is also important to emphasize that neither component alone is sufficient. Our experiments demonstrate that only the combination of both components leads to the best performance (as shown in Fig. 10(a) and Fig. 8). We further conducted a perturbation study to evaluate robustness under noisy depth estimation.  Specifically, we **add a perturbation study in Appendix J** that injects Gaussian noise into the depth map with variance is proportional to the depth range of each image to simulate realistic prediction errors:  $$\tilde{D}(x,y)=D(x,y)+\epsilon(x,y), \qquad\epsilon(x,y)\sim\mathcal{N}\left(0\; (\sigma\,\Delta D)^2\right),$$
>     where $D(x,y)$ is the original depth, $\Delta D = D_{\max}-D_{\min}$ denotes the depth range, and $\sigma$ is the noise factor. As shown in table below,  increasing noise levels do lead to a gradual decrease in editing accuracy; however, the performance remains stable under small and moderate noise, and only collapses when the injected noise becomes extremely large. These results show that GeoDrag remains reliable even with noisy depth predictions.
>
>     | $ \sigma $ | MD $\downarrow$ | DAI$_1$ $\downarrow$ | DAI$_{10}$ $\downarrow$ | DAI$_{20}$     $\downarrow$ | IF     $\uparrow$ |
>     |-----------|----------------|---------------------|-------------------------|    -------------------------|    ----------------|
>     | **Baseline** | **29.24** | **0.128** | **0.120** | **0.111** | **0.847** |
>     | $0.01$ | +0.66 | +0.000 | +0.000 | +0.000 | +0.00 |
>     | $0.05$ | +1.23 | +0.002 | +0.002 | +0.001 | +0.00 |
>     | $0.10$ | +2.85 | +0.004 | +0.003 | +0.003 | -0.002 |
>     | $0.50$ | +9.81 | +0.007 | +0.004 | +0.004 | -0.006 |
>
>    Additionally, we have expanded the failure case analysis to include more challenging scenarios, such as complex multi-object overlapping scenes and edits that require revealing previously unseen content. The results are shown in **the newly added Appendix K (Fig. 18)**. In these cases, the interpolation-based strategy conflicts with the need to synthesize entirely new structures rather than merely deform existing ones, leading to unsuccessful edits. A more advanced inpainting-based latent completion mechanism could potentially mitigate this issue, and we consider it a promising direction for future work.

---

> ### Author Response · Authors · 2025-11-22
> **Response to Reviewer KdHj (Part 2)**
>
> 4. Thank you for the constructive comment. To clarify the novelty of the plane-aware field beyond FastDrag, we provide both theoretical and empirical comparisons.
>     - **Theoretical comparison.** (1) FastDrag derives displacement using plane geometric constructions (e.g., similar triangles) to infer how each pixel should move. In contrast, our formulation is built on a ray–circle intersection interpretation, which leads to a simpler and fully vectorized closed-form expression that is computationally efficient and easier to integrate. (2) FastDrag computes displacement scaling based on the distance from the mask boundary, whereas we compute scaling by normalized distance between pixel p and the handle point. This formulation allows a natural introduction of the parameter $\beta$, enabling flexible editing strength. (3) While the underlying intuition of spatial decay is conceptually related to FastDrag, our plane-aware field is introduced explicitly to complement the geometry-aware field modeling, mitigating discontinuities and improving local controllability when geometry alone is insufficient.
>     - **Empirical comparison.** To further distinguish differences in practice, we benchmark the runtime of only the plane-aware field computation on *Other Object* subset from DRAGBENCH (35 samples). Our formulation consistently achieves lower computation time (**0.0109 seconds**) than FastDrag’s geometric construction (**0.4708 seconds**), which explains why our overall method does not incur significant runtime overhead despite introducing an additional field modeling component.
> 5. We thank the reviewer for raising this important point. Our choice of using a distance–only $\lambda$ is deliberate rather than a result of modeling constraints. The goal of $\lambda$ in Eq. (10) is to provide a user-controllable interpolation between global geometry-aware editing and local plane-aware refinement. In our design, the geometry-aware field governs global propagation consistent with 3D structure, whereas the plane-aware field acts as a local stabilizer that improves controllability around the handle region. Therefore, a distance-based $\lambda$ allows users to intuitively adjust the balance between these two behaviors without requiring an understanding of the scene’s depth distribution.
>
>     Coupling $\lambda$ with depth would reintroduce sensitivity to imperfect depth estimation, which can be unreliable in strong-perspective, specular, or textureless regions, and particularly discontinuous near object boundaries (see Fig. 17 in the Appendix). If $\lambda$ depends on depth, the fusion may oscillate between compensating and amplifying depth uncertainty,
>
>     **To further validate this design, we conducted two additional experiments.**
>     1. **Depth–based $\lambda$.**  We compute $\lambda$ based solely on the depth difference from the handle point. $$\lambda =\frac{|D - D_h|}{\max(|D - D_h|) + \epsilon},$$ where $D$ denotes the depth map and $D_h$ is the depth at the handle point. This configuration ensures that pixels with larger depth deviations from the handle point are assigned a higher geometry-aware weight, while near-depth pixels are encouraged to rely on the plane-aware refinement.
>     1. **Joint distance–depth $\lambda$.** To further investigate whether combining geometric and spatial cues could yield a better fusion mechanism, we replace the distance-only $\lambda$ with a joint formulation that simultaneously considers (1) pixel distance to the handle point and (2) depth discrepancy from the handle point. We first normalize the depth deviation relative to the handle depth: $$w_{\text{depth}} =\frac{|D - D_h|}{\max(|D - D_h|) + \epsilon},$$ and compute the spatial distance weight $$w_{\text{distance}} = P/(P+\lambda).$$ The two cues are then fused through a logistic gating mechanism: $$\lambda= \sigma\big(a \, w_{\text{distance}} + b \, w_{\text{depth}}\big),$$ where $a$ and $b$ control the relative contributions of spatial and depth cues, $\sigma$ is sigmoid function.  The corresponding results are shown in table below.

---

> ### Author Response · Authors · 2025-11-22
> **Response to Reviewer KdHj (Part3)**
>
> 5. **Continuation of Point 5**
>
>     | Fusion Strategy | MD $\downarrow$ | DAI$_1$ $\downarrow$ | DAI$_{10}$ $\downarrow$ | DAI$_{20}$ $\downarrow$ | IF $\uparrow$|
>     |- | - | - | - | - | - |
>     | **Distance only** | **29.24** | **0.128** | **0.120** | **0.111** | **0.847** |
>     | Deoth only | 35.26 |  0.162 |  0.155| 0.141|  0.857 |
>     | Joint distance–depth| 38.46 | 0.171 | 0.160 |0.144 | 0.848 |
>
>     These results are consistent with our analysis: introducing depth into $\lambda$ does not outperform the distance–only formulation. Both depth–only and joint distance–depth strategies lead to significantly worse editing accuracy (higher MD and DAI scores). It is also worth noting that both depth–only and joint strategies show a slightly higher IF score. However, this does not indicate better editing quality. IF measures fidelity to the original image, and when editing accuracy decreases, the system naturally stays closer to the input image, thus yielding a mechanically higher IF. Therefore, the increase in IF is a side effect of reduced editing effectiveness rather than evidence of improved fidelity–edit balance.
>
>     To summarize, depth already plays a central role in the geometry-aware field, while $\lambda$ is deliberately designed to control the influence range of this field rather than to encode additional cues. This separation ensures both intuitive user control and robust performance across imperfect depth conditions.
>
> 6. Thank you for pointing this out. We have now included comprehensive comparisons among GoodDrag (ICLR 2025), GeoDrag (original), and the optimized version of GeoDrag. The quantitative results are shown in the following table.
>
>    The optimized GeoDrag variant is obtained from the per-stage runtime analysis suggested by Reviewer VFmB. Following this suggestion, we conduct detailed per-stage profiling **(Appendix L, Table 5)**, which reveals that interpolation and latent relocation are the two major bottlenecks, aside from DDIM inversion and sampling. We therefore focus on improving the efficiency of these two components. Specifically, (1) we fully vectorize latent relocation to remove unnecessary per-point loops, and (2) we replace the original dynamic BNNI interpolation (which requires sequential updates) with a static 4-neighbor interpolation (up / down / left / right), which can be parallelized. This change makes interpolation and latent relocation orders of magnitude faster (e.g., interpolation time drops from 0.2247 s to 0.0014 s, and latent relocation from 0.2714 s to 0.0009 s per edit). Interestingly, this simplified interpolation not only reduces runtime but also slightly improves quantitative performance (e.g., MD decreases from 29.24 to 27.84), possibly due to remove iterative BNNI updates avoids error accumulation while preserving the essential semantic structure in the latent space.
>
>    While GoodDrag achieves lower MD and higher IF score, this comes at the cost of significantly slower runtime: 71.34 seconds per point, compared to 3.95 seconds for GeoDrag (original) and 1.94 seconds for GeoDrag (optimized), making GoodDrag roughly $18×$ and $36×$ slower, respectively. GeoDrag is designed for real-time, geometry-aware interactive editing rather than iterative optimization, enabling accurate multi-point and geometry-aware deformation edits that remain stable and visually consistent. We believe this represents a practical and complementary trade-off between edit precision and interactive usability. We also provide qualitative comparisons with GoodDrag in **the newly added Appendix M Fig. 20**. The results show that GeoDrag produces visually comparable or even better editing outcomes, while maintaining structural consistency and achieving much higher efficiency.
>
>     | Approach          | MD $\downarrow$ | DAI$_1$ $\downarrow$ | DAI$_{10}$ $\downarrow$ | DAI$_{20}$ $\downarrow$ | IF $\uparrow$ |  Time (s) |
>     |-------------------|-----------------|----------------------|--------------------------|--------------------------|---------------| ---------------|
>     | GoodDrag          |   26.65    |        0.153     |      0.133      |       0.124       |   0.871    | 71.34 |
>     | GeoDrag (optimized) | 27.84 | 0.133 | 0.121 | 0.111 | 0.847 | 1.94 |
>     | GeoDrag (original) | 29.24 | 0.128 | 0.120 | 0.111 | 0.847 | 3.95 |

---

### Official Review · Reviewer_VFmB · 2025-11-01

**Soundness:** 3
**Presentation:** 3
**Contribution:** 3
**Rating:** 6
**Confidence:** 3

**Summary:**

This paper proposes GeoDrag, a drag-based image editor that injects 3D geometry into 2D dragging. The authors derive a geometry-aware influence field. Specifically, per-pixel motion scales with the relative depth to a handle point, so nearer pixels move more than farther ones, reducing perspective artifacts during editing. For multi-handle cases, they introduce a conflict-free hard partition so each pixel is driven by exactly one handle, avoiding direction cancellation. On DragBench, GeoDrag achieves the best edit precision (lowest MD/DAI) with competitive speed/memory.

**Strengths:**

* The derivation from 3D displacement to 2D motion motivates a depth-modulated field that is simple, interpretable, and implementation-light. The proposed approach effectively reduces perspective-induced tearing during large drags.
* On DragBench, GeoDrag attains best MD and best DAI1/10/20 with a one-step pipeline and 3.95s per point / 5.44 GB peak memory, competitive with FastDrag’s 3.23s / 5.85 GB and faster than LoRA-prep methods (∼1-min prep). Compared with baselines, for which LoRA-prep is often necessary, the proposed method shows better generalizability.
* The partitioning successfully creates predictable, non-interfering edits. Ablations show it outperforms soft blends like Directly-Add, Pixel-Distance, and Drag-Magnitude across MD/DAI/IF (Fig. 10b)

**Weaknesses:**

* The simplification that neglects motion along the optical axis to obtain is reasonable for small motions, but the paper does not quantify robustness to depth noise (from monocular estimation), likely failure modes for strong perspective or specular/textureless regions. A perturbation study of these factors would better support claims of geometric stability. Additional failure cases will also help to understand the limitations of the work.
* The runtime table shows FastDrag is still faster per point (3.23 s) than GeoDrag (3.95 s) on the same GPU, and the manuscript does not provide a full stack breakdown (depth inference, partitioning, field fusion, masked DDIM) to explain where time goes or how to optimize it. A consolidated per-stage timing would strengthen the efficiency story.
* Although the proposed method is claimed to work better for long-range drag operation, the qualitative results show that it still has problems where the handle points are not actually moved to the target points (Figure 5 Ours, there are always small distances). However, such a case is usually not seen in copy-paste methods like SDEDrag.
* The geometry-aware influence field is derived under small-motion, locally smooth assumptions and works best when pixels “move with” nearby handles. However, edits that require self-occlusion swaps or reveal previously unseen regions may lead to problems.

**Questions:**

* In practice, one common idea is that LoRA should improve the performance in drag editing. However, Table 2 suggests that LoRA actually hurts the performance for most metrics. Is there any further explanation or examples?

---

> ### Author Response · Authors · 2025-11-22
> **Response to Reviewer VFmB (Part 1)**
>
> Thank you for your insightful comments! In the following, we provide our point-by-point response and hope our response helps address your concerns. We also look forward to the subsequent discussion which may further help solve the current issues.
>
> 1. **Robustness to depth noise.** We appreciate your suggestion. Indeed, in certain challenging scenarios such as specular or textureless regions, monocular depth estimation may be unreliable. For example, in overly smooth areas (e.g. walls), the predicted depth map often degenerates into a nearly uniform plane, providing little or no meaningful geometric gradient. In such cases, the geometry-aware displacement field alone cannot provide sufficiently reliable deformation guidance.
>
>     However, our method remains robust in these situations, benefiting from the **Spatial Plane Modulation**. This modulation computes the displacement field purely based on pixel distances in the spatial plane and does not rely on semantic cues or depth gradients. This makes it naturally insensitive to depth failures and capable of producing stable deformations even when geometry signals are ambiguous.
>
>     - **Challenging depth scenarios.** To assess robustness to depth noise, we carefully selected several representative examples that match the challenging scenarios, including transport-specular regions, textureless regions, and strong perspective view. For each case, we present results under (1) geometry-aware displacement field alone, and (2) the combined displacement field used in our method. The results are **added in Appendix J (Fig. 17)**. These examples reveal different depth-prediction failures: transparent–specular surfaces collapse depth into a coarse foreground/background split, textureless regions provide almost no usable depth gradient, and strong perspective leads to large depth discontinuities. In all cases, geometry-only guidance becomes unstable due to the unreliable depth cues. By incorporating spatial plane modulation, our method compensates for these issues with a depth-independent, spatially smooth prior, yielding more stable deformations.
>
>     - **Injected depth noise.** Moreover, we conducted **an additional perturbation study in Appendix J (Table 4)** to assess the robustness of GeoDrag under noisy depth predictions. To simulate realistic depth prediction error, we inject Gaussian noise whose variance is proportional to the depth range of each image:
>     $$\tilde{D}(x,y)=D(x,y)+\epsilon(x,y), \qquad\epsilon(x,y)\sim\mathcal{N}\left(0\; (\sigma\,\Delta D)^2\right),$$
>     where $D(x,y)$ is the original depth, $\Delta D = D_{\max}-D_{\min}$ denotes the depth range, and $\sigma$ is the noise factor. The quantitative results are reported in **Appendix J (Table 4) and summarized below**. Overall, noise in the depth map does influence the dragging accuracy, the larger the noise, the more performance tends to drop. However, the results also show that GeoDrag has a certain degree of robustness: under small and moderate noise levels, the metrics remain relatively stable instead of collapsing immediately, and only when the noise becomes extremely strong (e.g., $\sigma = 0.5$) do we observe a noticeable decline, indicating that GeoDrag can still function reliably even when the depth prediction is not very precise.
>
>
>         | $ \sigma $ | MD $\downarrow$ | DAI$_1$ $\downarrow$ | DAI$_{10}$ $\downarrow$ | DAI$_{20}$     $\downarrow$ | IF     $\uparrow$ |
>         |-----------|----------------|---------------------|-------------------------|    -------------------------|    ----------------|
>         | **Baseline** | **29.24** | **0.128** | **0.120** | **0.111** | **0.847** |
>         | $0.01$ | +0.66 | +0.000 | +0.000 | +0.000 | +0.00 |
>         | $0.05$ | +1.23 | +0.002 | +0.002 | +0.001 | +0.00 |
>         | $0.10$ | +2.85 | +0.004 | +0.003 | +0.003 | -0.002 |
>         | $0.50$ | +9.81 | +0.007 | +0.004 | +0.004 | -0.006 |

---

> ### Author Response · Authors · 2025-11-22
> **Response to Reviewer VFmB (Part 2)**
>
> 2. **Per-stage timing.** We agree that providing a per-stage runtime breakdown can help clarify efficiency differences. Thus, we provide **additional per-stage runtime results in Appendix L (Table 5) and reproduced below**, covering depth inference, partitioning, geometry- and plane-aware field computation, latent relocation, interpolation, and DDIM inversion and sampling. Note that the model loading time and several preprocessing operations are excluded from this profiling, which results in the total runtime being slightly lower than the time reported in Table 1. We used the *Other Object* subset from DRAGBENCH (35 samples).
>
>     | Stage | DDIM inversion | DDIM sampling | Interpolation | Depth prediction | Plane-aware field computation | Latent relocation | Partition | Geometry-aware field computing |
>     |-|-|-|-|-|-|-|-|-|
>     | Time (s) | 1.2721 | 1.2416 | 0.2247 | 0.0995 |  0.0109 | 0.2714 | 0.0006 | 0.0003 |
>
>     The results show that, apart from DDIM inversion and DDIM sampling, the most time-consuming stages are interpolation and latent relocation. Therefore, we focuse our efficiency improvements specifically on these two modules.
>    - **Latent relocation.** We observed that its implementation can be fully vectorized, allowing us to remove unnecessary per-point loops.
>    - **Interpolation.** The original BNNI strategy is dynamic—each location depends on updated values of other locations, which forces a sequential scan. We simplify this process by using a static interpolation rule: each location is interpolated only once based on its nearest four spatial neighbors (up / down / left / right), without dynamic updates. This enables the interpolation step to be parallelized and run significantly faster.
>
>     Surprisingly, this simplified interpolation not only reduces runtime but also improves performance. We believe this is because removing the iterative BNNI updates prevents error accumulation, while the static strategy still retains the essential semantic structure in the latent space. The quantitative comparison is reported in the table below.
>
>     | Model | MD $\downarrow$ | DAI$_1$ $\downarrow$ | DAI$_{10}$ $\downarrow$ | DAI$_{20}$ $\downarrow$ | IF $\uparrow$ |     Interpolation time | Latent relocation time |
>     |-|-|-|-|-|-|-|-|
>     | GeoDrag (dynamic interpolation / original) | 29.24 | **0.128** | **0.120** | 0.111 | 0.847 | 0.2247 | 0.2714 |
>     | GeoDrag (static interpolation / optimized) | **27.84** | 0.133 | 0.121 | 0.111 | 0.847 | **0.0014** | **0.0009** |
>
> 3. **Handle–target gap.** The slight gap between the handle and target points mainly comes from the fact that the drag operation is performed in the latent space rather than directly on image pixels. When converting drag points from the pixel domain to the diffusion latent (e.g., from $512 \times 512$ pixels to $64 \times 64$ latent), the coordinates must be downscaled accordingly. This conversion inevitably reduces the effective displacement magnitude. To mitigate this effect, we introduce a manually defined scaling factor that rescales the displacement field after latent coordinate conversion, compensating for the attenuation introduced during downsampling. The **additional results are illustrated in Appendix M (Fig. 19)**, adjusting this scaling factor indeed influences the alignment between the handle and target points. A moderate increase in scale ($1.2-1.3$) improves the editing accuracy by compensating for the displacement attenuation caused by downsampling. However, excessively amplifying the scale overcorrects the motion and leads to geometric distortion and artifacts. This suggests that an appropriate scale strikes a balance between accurate point alignment and preserving realistic deformation.
>
>    Although the drag points may not be perfectly aligned with the target points in Fig. 5, our method consistently achieves the best results in complex scenarios such as multi-point editing and geometry-aware deformation. These observations confirm the practical effectiveness and advantage of our approach, especially in challenging editing setups that cannot be handled reliably by copy-paste approaches like SDEDrag.
>
>    In addition, our method delivers better performance than SDEDrag. We select three representative samples from Fig. 5 for a direct comparison and find that our approach achieves noticeably higher editing accuracy under default setting. As shown in **additional Appendix N (Fig. 20)**, SDEDrag struggles to generate sufficient deformations, while GeoDrag produces accurate and visually consistent results.

---

> ### Author Response · Authors · 2025-11-22
> **Response to Reviewer VFmB (Part 3)**
>
> 4. **Failed cases.** We agree with you that edits involving self-occlusion swaps or revealing previously unseen regions can lead to failure cases. In such cases, the current interpolation-based strategy inevitably conflicts with the goal of synthesizing new structures rather than deforming existing ones. To make this limitation explicit, we have added failure examples exposing previously unseen content in **Appendix K (Fig. 18)**. The example on the left (Fig. 18) involves a complex scene with multiple overlapping objects. Although the editing direction is partially correct, the newly exposed regions appear distorted because interpolation reconstructs unseen content by sampling nearby latent features, and dense object overlaps introduce semantic ambiguity. The example on the right (Fig. 18) need to generate a region that is completely invisible in the original image; without any valid correspondence in the latent space, interpolation fails to provide structural cues, resulting in editing failure. A possible way to mitigate this limitation is to replace interpolation with a more advanced inpainting-based latent completion strategy, which represents a promising direction for future work.
> 5. **Effect of LoRA.** We thank you for raising this point. When conducting this experiment, we also expected LoRA to consistently enhance drag-editing performance, and therefore we carried out a thorough qualitative and quantitative evaluation. Interestingly, the results show that LoRA does not universally improve performance. We analyze this phenomenon and find that **LoRA, being fine-tuned on the original image, enforces strong identity preservation, which enhances fidelity but inevitably reduces editing freedom**. As shown in Fig. 24 of the Appendix, LoRA is helpful when the goal is identity-preserving editing (e.g., rows 4, 7, and 8), but it suppresses structural changes and leads to incomplete edits in scenarios requiring more deformation freedom (e.g., rows 3 and 9). This trade-off is also reflected in Table 2: LoRA improves the IF metric, demonstrating higher fidelity to the original image, but slightly decreases the metrics related to editability. Therefore, whether to enable LoRA depends on the desired objective: when identity consistency is prioritized, LoRA is beneficial; when higher editing freedom is required, using the backbone without LoRA produces better results.

---

### Author Response · Authors · 2025-11-22
**Global Response to Reviewers**

We thank all reviewers for their thoughtful, constructive, and encouraging feedback. We are encouraged that reviewers recognized the novelty and significance of incorporating 3D geometry into drag-based image editing (Reviewer VFmB, KdHj, 2Gtw, and m8ds). We are glad that reviewers recognized our method for achieving precise editing while maintaining competitive runtime and memory performance (Reviewer VFmB, 2Gtw, and m8ds), and praised the efficiency and usability of our framework (Reviewer KdHj, 2Gtw, and m8ds).

We sincerely appreciate the reviewers for their valuable insights. Our work has benefited significantly from this feedback. Below we summarize the revisions and additional analyses added to the manuscript (highlighted in blue in the revised PDF):
1. **Robustness to Depth Map.** To address the concerns of Reviewers VFmB, KdHj, and m8ds, we conducted an extensive robustness analysis on depth estimation in Appendix J (Fig. 17 and Table 4). Specifically, we evaluated performance under challenging scenarios, including transport-specular regions, textureless regions, and strong perspective view and further performed a perturbation study with synthetic depth noise. The results demonstrate that our method remains stable and robust even under imperfect depth predictions.
2. **Per-Stage Runtime.** In response to Reviewer VFmB, we added a per-stage runtime breakdown of the entire pipeline, covering depth inference, partitioning, geometry- and plane-aware field computation, latent relocation, interpolation, and DDIM inversion and sampling. The analysis (Appendix L Table 5) shows that the majority of latency originates from the frozen diffusion backbone rather than our additional modules. We further provide an optimized configuration that reduces the remaining overhead, demonstrating that GeoDrag achieves interactive performance without compromising its dragging accuracy.
3. **Effect of the Scaling Factor.** To clarify the small residual gap between handle and target points noted by the reviewers VFmB, we added an explanation of the latent–pixel coordinate conversion and its influence on displacement magnitude. We further visualize (Appendix L Fig. 14) how applying a scaling factor compensates for the attenuation and improves alignment, and show that a moderate value ($1.2–1.3$) offers an effective balance between point accuracy and geometric consistency.
4. **Additional Comparisons.** As requested by Reviewer VFmB and 2Gtw, we added qualitative comparisons with SDEDrag and GoodDrag in Appendix N (Fig. 20). The results show that GeoDrag delivers comparable or better editing quality, while being significantly more efficient since it requires neither LoRA fine-tuning nor multi-step optimization.
5. **Additional Visual Result.** In response to Reviewer 2Gtw, we added supplementary visual results in Appendix C.3 (Fig. 14) demonstrating how adjusting the scaling parameter $\eta$ enhances texture fidelity and preserves visual identity during editing.
6. **Failure Cases Analysis.** In response to Reviewer KdHj, we added representative failure cases in Appendix K (Fig. 18), including an overlapping-object scene and a case requiring unseen content generation.

We hope that the above revisions and the following point-by-point clarifications fully address the reviewers’ questions. If our responses are satisfactory, we would be grateful if you could consider updating your evaluations accordingly.

We sincerely appreciate the reviewers’ time and constructive feedback, and we warmly welcome any further discussions.

---

### Author Response · Authors · 2025-11-30
**General Response to AC**

Dear ACs,

We sincerely appreciate your time and effort in overseeing the review process during this unusual cycle. For your convenience, we summarize how each reviewer’s comments are addressed in the rebuttal and revised manuscript. Although no follow-up messages are posted, every comment is responded to carefully. The points below are meant to help the ACs interpret the current reviews together with the revised version.
# **Brief Summary of Reviewer Comments**
We thank all reviewers for their feedback. Three reviewers (VFmB, KdHj, 2Gtw) recommend borderline acceptance, while reviewer m8ds gives a score of 4 (confidence: 2). Several of reviewer m8ds’s concerns seem to stem from misunderstandings of the original submission:
- **Ablation insufficiency.** We would like to clarify that this concern stems from a misunderstanding rather than from a lack of experiments. The ablations were already included in the initial submission (Figure 8 for qualitative and Figure 10(a) for quantitative results).
- **Efficiency discussion shallow.** We respectfully clarify that runtime and memory comparisons were already provided in the initial submission (Figure 2(c), Table 1, and Table 3).
- **Applicability scope.** We respectfully clarify that this concern arises from a misunderstanding between the reviewer’s expectation and the intended scope of our method. Video editing and image editing, diffusion and non-diffusion architectures are fundamentally different problem settings and fall outside the scope of this work by design, not as a limitation.

Additionally, we addressed the remaining concerns by adding robustness experiments in challenging scenes, clarifying the theoretical grounding of latent-space point dragging with citations, and clarifying the displacement-field figure with a high-resolution version on the project page.
# **Rebuttal Summary**
To support quick assessment, we briefly summarize the main updates and clarifications provided in our rebuttal:

**1. Robustness to Depth Map.** We added a robustness study in Appendix J, showing the method remains stable under imperfect depth. *See Response to Reviewer VFmB Part 1, Point 1.*

**2. Per-stage timing.** We added a per-stage runtime breakdown in Appendix L, showing that interpolation and latent relocation are the main overhead; optimizing them improves both speed and performance. *See Response to Reviewer VFmB Part 2, Point 2.*

**3. Handle–target gap.** We conducted additional experiments (Appendix M and N), showing that applying an appropriate scaling factor effectively reduces the gap, and our method achieves higher editing precision than SDEDrag. *See Response to Reviewer VFmB Part 2, Point 3.*

**4. Failed cases.** We added representative failure cases in Appendix K to make the method’s scope explicit. *See Response to Reviewer VFmB Part 3, Point 4.*

**5. Novelty Beyond FastDrag.** We clarified the novelty of the plane-aware field through both theory and experiments. Our ray–circle formulation is simpler, fully vectorized, and much faster (0.0109s (ours) vs. 0.4708s (FastDrag)). *See Response to Reviewer KdHj Part 2, Point 4.*

**6. Fusion Parameter $\lambda$.** We evaluated whether $\lambda$ should incorporate depth. Experiments show that our original design is best. *See Response to Reviewer KdHj Part 2, Point 5, and Part 3 continuation of Point 5.*

**7. Comparison with GoodDrag.** We benchmarked GoodDrag and GeoDrag; GeoDrag achieves comparable or better visual quality and up to 36× faster speed, enabling real-time interactive editing. *See Response to Reviewer KdHj Part 3, Point 6.*

**8. Texture preservation.** We added experiments in Appendix C.3 (Fig. 14) demonstrating non-zero $\eta$ produces more faithful texture preservation. *See Response to Reviewer 2Gtw Part 1, Point 1.*

### **Other questions:**
1. We clarified when LoRA helps and when it reduces editing freedom, addressing the question of Reviewer VFmB. *See Response to Reviewer VFmB Part 3, Point 5.*
2. We answered the reviewer’s question on foreground–background co-movement by explaining why it can occur in depth-entangled scenes. *See Response to Reviewer 2Gtw Part 1, Point 2.*
3. We explained why a 2.5D formulation is used for interactive performance and why full 3D motion is computationally incompatible with real-time dragging. *See Response to Reviewer 2Gtw Part 1, Point 3.*
4. We agreed that integrating inpainting with geometry-aware deformation is promising and complementary. *See Response to Reviewer 2Gtw Part 2, Point 4.*
5. We clarified that our method already runs with monocular depth. *See Response to Reviewer m8ds Part 3, Point 7.*
6. We clarified that the diffusion model remains frozen; the method is fully training-free. *See Response to Reviewer m8ds Part 3, Point 8.*

We fully respect the decision to revert reviews for fairness and provide this note only to help the ACs interpret the present reviews together with the revised manuscript.

Best Regards,

---

### Meta-Review · Area_Chair_4KBC · 2026-01-06

**Summary:**

The submission received mixed evaluations, with three positive scores and one negative score.

Reviewer m8ds, who provided the sole negative rating, raises three primary concerns: (1) the method relies on depth input, which may be susceptible to noise; (2) the ablation study and efficiency analysis are insufficient; and (3) the method does not generalize to video settings or non-diffusion models.

The remaining three reviewers are generally positive. Their main concerns include: (1) the lack of a quantitative analysis of robustness to depth noise; (2) the need for clearer runtime breakdowns; (3) clarification of the novelty against FastDrag; and (4) additional experimental analyses and comparisons.

**Reviewer Concerns:**

The Area Chair has carefully reviewed the rebuttal and believes that the authors have adequately addressed the reviewers’ concerns in a point-by-point manner.

**Reviewer Scores:**

Reviewers VFmB, KdHj, and 2Gtw maintain their original positive ratings. Reviewer m8ds is inclined to increase the rating from negative to positive.

---

### Decision · Program_Chairs · 2026-01-26

Accept (Poster)